# Iron Isotope Fractionation during Skarn Cu-Fe Mineralization

Song Xue [1,*], Yaoling Niu [1,2,3,*], Yanhong Chen [1], Yining Shi [1], Boyang Xia [1], Peiyao Wang [3,4,5], Hongmei Gong [3,4,5], Xiaohong Wang [3,4,5] and Meng Duan [3,4,5]

1. School of Earth Sciences and Resources, China University of Geosciences, Beijing 100083, China; chenyh@cugb.edu.cn (Y.C.); Yining_Shi@outlook.com (Y.S.); Xia_Bo_Yang@163.com (B.X.)
2. Department of Earth Sciences, Durham University, Durham DH1 3LE, UK
3. Laboratory for Marine Geology, Qingdao National Laboratory for Marine Science and Technology, Qingdao 266061, China; peiyao.wang@qdio.ac.cn (P.W.); gonghm@qdio.ac.cn (H.G.); wangxiaohong@qdio.ac.cn (X.W.); m.duan@foxmail.com (M.D.)
4. Institute of Oceanology, Chinese Academy of Sciences, Qingdao 266071, China
5. Center for Ocean Mega-Science, Chinese Academy of Sciences, 7 Nanhai Road, Qingdao 266071, China
* Correspondence: song.x@cugb.edu.cn (S.X.); yaoling.niu@durham.ac.uk (Y.N.)

**Abstract:** Fe isotopes have been applied to the petrogenesis of ore deposits. However, the behavior of iron isotopes in the mineralization of porphyry-skarn deposits is still poorly understood. In this study, we report the Fe isotopes of ore mineral separations (magnetite, pyrite, chalcopyrite and pyrrhotite) from two different skarn deposits, i.e., the Tonglvshan Cu-Fe skarn deposit developed in an oxidized hydrothermal system and the Anqing Cu skarn deposit developed in a reduced hydrothermal system. In both deposits, the Fe isotopes of calculated equilibrium fluids are lighter than those of the intrusions responsible for the skarn and porphyry mineralization, corroborating the "light-Fe fluid" hypothesis. Interestingly, chalcopyrite in the oxidized-Tonglvshan skarn deposit has lighter Fe than chalcopyrite in the reduced-Anqing skarn deposit, which is best understood as the result of the prior precipitation of magnetite (heavy Fe) from the ore fluid in the oxidized-Tonglvshan systems and the prior precipitation of pyrrhotite (light Fe) from the ore fluid in the reduced-Anqing system. The $\delta^{56}Fe$ for pyrite shows an inverse correlation with $\delta^{56}Fe$ of magnetite in the Tonglvshan. In both deposits, the Fe isotope fractionation between chalcopyrite and pyrite is offset from the equilibrium line at 350 °C and lies between the FeS-chalcopyrite equilibrium line and pyrite-chalcopyrite equilibrium line at 350 °C. These observations are consistent with the FeS pathway towards pyrite formation. That is, Fe isotopes fractionation during pyrite formation depends on a path from the initial FeS-fluid equilibrium towards the pyrite-fluid equilibrium due to the increasing extent of Fe isotopic exchange with fluids. This finding, together with the data from other deposits, allows us to propose that the pathway effect of pyrite formation in the Porphyry-skarn deposit mineralization is the dominant mechanism that controls Fe isotope characteristics.

**Keywords:** porphyry-skarn deposits; Fe isotopes; isotopic fractionation; pathway effects; light-Fe fluid; redox state

## 1. Introduction

Iron isotopes, as a possible tracer of geological processes, have attracted much attention in recent years [1]. Current studies have reported that there are significant variations in Fe isotope ratios in varying rocks and solid earth geochemical reservoirs [2–5] as a consequence of mantle sources [6–10], magma differentiation [11–15] and chemical and thermal diffusion on microscopic scales [16,17].

Traditionally, light stable isotopes (e.g., H, O, S) have been used to study sources and processes of mineralization [18–20], contributing to the understanding of the petrogenesis of ore deposits, but cannot provide more direct information since these elements are not ore-forming metals [21,22].

Hence, Fe isotopes and other metal stable isotopes have been used to study the ore genesis due to the advancement of analytical methods on stable isotopes of ore metals (e.g., Fe, Cu, Zr, Ni) [21–39]. However, two important problems remain concerning magmatic-hydrothermal deposits (e.g., porphyry- and skarn-type deposits): (1) Fe isotope fractionation during fluid exsolution [23–25,32,33]; (2) Fe isotope behavior in fluid-mineral systems (e.g., mineral-fluid fractionation and between mineral fractionation). Regarding question (1), there are mainly two views. One is that the exsolved magmatic-hydrothermal fluid has a light Fe isotopic composition [40,41] in the range of $\delta^{56}Fe = -0.39‰$ and $-0.05‰$. The other [32] is that the redox state actually governs the Fe isotope composition of the exsolved fluid. For example, oxidized magmas crystallize magmatic magnetite with heavy Fe, resulting in a melt having lighter Fe and hence a lighter Fe magmatic-hydrothermal fluid. On the other hand, reduced magmas crystallize ferrous minerals (e.g., pyroxenes and ilmenite) with light Fe, leading to a melt having heavier Fe and hence a heavier Fe magmatic-hydrothermal fluid. For question (2), theoretical calculations and related experimental studies have provided equilibrium fractionation factors for minerals and fluids [42–46]. However, for natural samples, it is not straightforward to determine whether the co-existing minerals and fluid may have achieved Fe isotope equilibrium; hence kinetic Fe isotope fractionation during mineralization is thought to be important [44,47]. Alternatively, the pathway of pyrite formation may control pyrite-fluid Fe isotope compositions [44,48–50], which in turn affects the Fe isotope composition of fluids and subsequent minerals [31].

In this study, we compare two different skarn ore deposits, the "oxidized" Tonglvshan skarn Cu-Fe-Au deposit and the "reduced" Anqing skarn Cu deposit, in the same metallogenic belt with the attempt to better understand the problem (1), which also helps understand problem (2). We analyzed Fe isotope compositions of major ore minerals magnetite, pyrite, chalcopyrite and pyrrhotite. The data support that the pathway effect of pyrite formation is a common mechanism of Fe isotope fractionation that controls Fe isotope compositions of pyrite, fluids and co-existing minerals in the mineralization of skarn deposits.

## 2. Geology and Samples

The Middle-Lower Yangtze River (MLYR) metallogenic belt (Figure 1) is a famous Cu-Fe-Au porphyry-skarn province in China, from east to west, consisting of Ningzhen, Ningwu, Luzong, Tongling, Anqing-Guichi, Jiurui and Edong ore districts, all of which are associated with late Mesozoic granitoid magmatism [51,52]. We collected 15 skarn ore samples from the Tonglvshan skarn deposit, which lies in the Edong ore district, and the Anqing skarn deposit from the Anqing-Guichi ore districts.

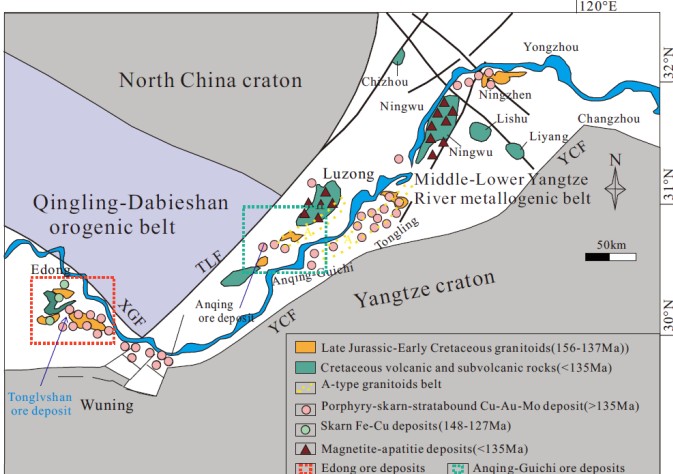

**Figure 1.** Geological map of the Middle-Lower Yangtze River Valley metallogenic belt in eastern continental China (modified from Ref. [52]).

## 2.1. Tonglvshan Cu-Fe-Au Skarn Deposit

The Tonglvshan Cu-Fe-Au deposit is associated with an Yangxin quartz diorite stock intruding the dolomitic limestones of the Lower Triassic Daye Formation, resulting in skarn mineralization (Figure 2) [53]. Previous studies report that this deposit contains economic metals of 1.08 Mt Cu (1.78% Cu), 60 Mt Fe (41% Fe), 70 t Au (0.38 g/t Au), and 508 t Ag [54]. A recent dating study [53], using the Laser ablation ICP-MS titanite U-Th-Pb method, shows evidence of two independent hydrothermal events (~136 Ma and 121 Ma). A more detailed study [55] suggested the ore-forming fluid to be of magmatic-hydrothermal origin. Detailed fluid inclusion work suggested the magnetite deposition temperature of 405–567 °C and quartz-sulfide mineralization temperature of 240–350 °C [55].

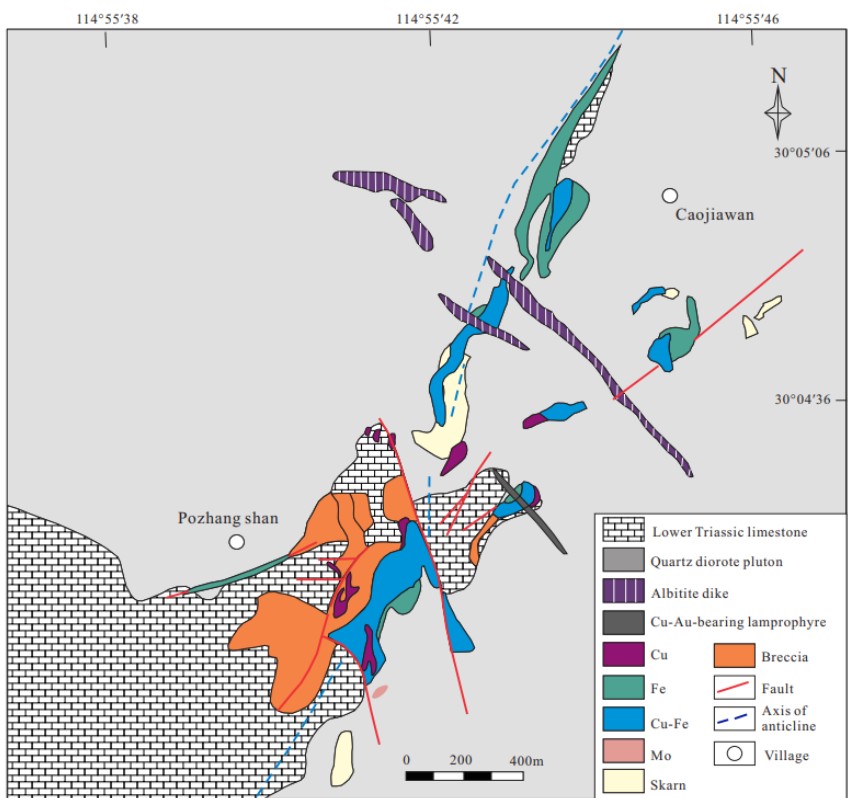

**Figure 2.** Geological map of the Tonglvshan Cu-Fe-Au skarn deposit in eastern continental China (modified from Ref. [53]). Cu, Fe, Cu-Fe, Mo: skarn ore bodies.

Magnetite, pyrite and chalcopyrite are the three main ore minerals in the Tonglvshan skarn deposit (see Table 1 for petrography), sharing a similar mineral assemblage to oxidized porphyry-skarn deposits (Figure 3) [56]. The subhedral to euhedral magnetite crystals are characterized by abundant porous pits (Figure 3a,b,d–f) caused by the dissolution–reprecipitation process (DRP) [57–59]. Some authors [57] suggest that the DRP may be important in the formation of hydrothermal magnetite. The magnetite is mostly replaced and cut by sulfides, suggesting that the pyrite and chalcopyrite may form later than magnetite (Figure 3a,b,d–f).Thereby, the paragenetic sequence may be as follows: magnetite → pyrite/chalcopyrite.

**Table 1.** Petrography of the Tonglvshan skarn ore samples.

| Skarn Ore | Sample | Description |
|---|---|---|
| Ore body III | TLS3-1 | granular magnetite (~200 μm), subhedral pyrite (~2 mm), anhedral chalcopyrite (~300 μm) (Figure 3a). |
| | TLS3-3 | magnetite, pyrite, chalcopyrite. |
| | TLS3-5 | anhedral magnetite (~1 mm), euhedral pyrite (~1 mm), subhedral chalcopyrite (~3 mm) (Figure 3b). |
| | TLS3-6 | magnetite, pyrite, chalcopyrite. |
| | TLS3-9 | acicular hematite (~1 mm), euhedral pyrite (~3 mm) (Figure 3c). |
| | TLS3-11 | anhedral magnetite (~1 mm), subhedral pyrite (~3 mm), subhedral chalcopyrite (~2 mm) (Figure 3d). |
| | TLS3-12 | disseminated magnetite (~100 μm), euhedral pyrite (~2.5 mm), anhedral chalcopyrite (~1 mm) (Figure 3e). |
| Ore body IV | TLS4-7 | disseminated magnetite (~200 μm), anhedral pyrite (~1 mm), anhedral chalcopyrite (~100 μm) (Figure 3f). |

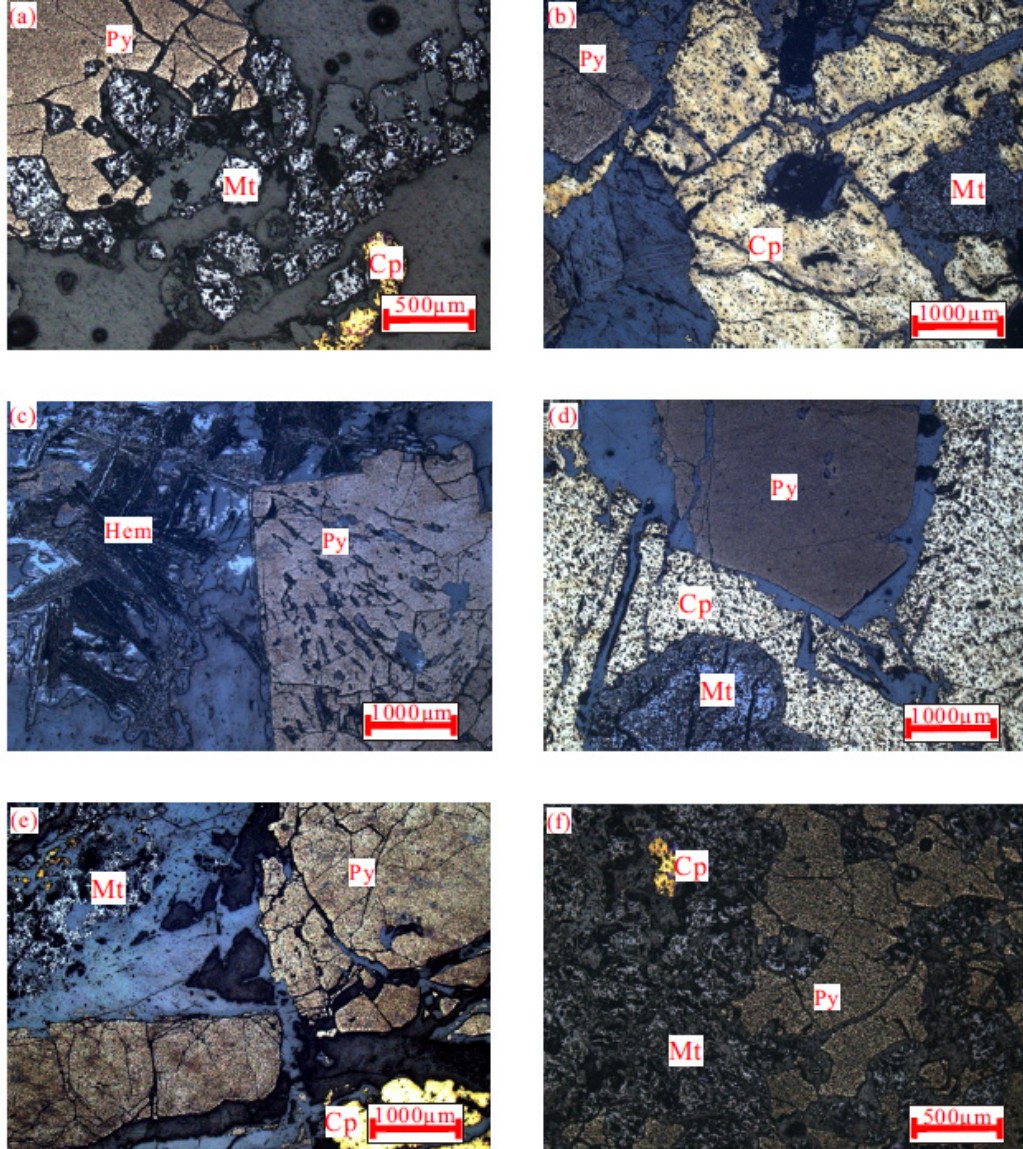

**Figure 3.** Photomicrographs of selected hydrothermal mineral assemblages from the Tonglvshan deposit. Mt = magnetite; Py = pyrite; Cp = chalcopyrite; Bn = Bornite; Hem = hematite. (**a**) skarn ore sample(TLS3-1) with granular magnetite, subhedral pyrite and anhedral chalcopyrite. (**b**) skarn ore sample(TLS3-5) with anhedral magnetite, euhedral pyrite and subhedral chalcopyrite. (**c**) skarn ore sample(TLS3-9) with acicular hematite and euhedral pyrite. (**d**) skarn ore sample(TLS3-11) with anhedral magnetite, subhedral pyrite and subhedral chalcopyrite.(**e**) skarn ore sample(TLS3-12) with disseminated magnetite, euhedral pyrite and anhedral chalcopyrite. (**f**) skarn ore sample(TLS4-7) with disseminated magnetite, anhedral pyrite and anhedral chalcopyrite.

## 2.2. Anqing Cu skarn Deposit

The Anqing Cu deposit is a typical skarn ore deposit in the Yueshan metallogenic belt [60], located within the contact between the dioritic intrusions and the Low Triassic Yueshan Formation (limestone) (Figure 4). A previous study divided the mineralization into four stages: (I) garnet-diopside skarn stage, (II) oxides stage, (III) quartz-sulfide stage and (IV) quartz-carbonate stage [60]. The fluid inclusion work suggests the temperature of 200–375 °C at the quartz-sulfide stage [60].

The main mineral assembles in the Anqing Cu deposit are pyrite, chalcopyrite and pyrrhotite, similar to reduced porphyry-skarn deposit (Figure 5; see Table 2 for petrography) [56]. The anhedral and subhedral pyrrhotite is replaced and cut by anhedral chalcopyrite, indicating that chalcopyrite formed later than pyrrhotite (Figure 5a–d). Pyrite is subhedral and replaces the disseminated pyrrhotite, suggesting that the pyrite formed later than pyrrhotite (Figure 5f). Hence, the paragenetic sequence may be as follows: pyrrhotite → pyrite/chalcopyrite.

**Table 2.** Petrography of the Anqing skarn ore samples.

| Skarn Ore | Sample | Description |
|---|---|---|
| Anqing skarn deposit | AQ-1 | disseminated chalcopyrite (~200 μm), disseminated pyrrhotite (~200 μm) (Figure 5a). |
| | AQ-2 | chalcopyrite, pyrrhotite, pyrite. |
| | AQ-3 | anhedral chalcopyrite (~300 μm), anhedral pyrrhotite (~300 μm), anbhedral pyrite (~500 μm) (Figure 5b). |
| | AQ-5 | anhedral chalcopyrite (~300 μm), anhedral pyrrhotite (~200 μm) (Figure 5c). |
| | AQ-7 | anhedral chalcopyrite (~100 μm), subhedral pyrrhotite (~1 mm) (Figure 5d). |
| | AQ-8 | subhedral pyrite (~1 mm), disseminated chalcopyrite (~100 μm) (Figure 5e). |
| | AQ-9 | subhedral pyrite (~1 mm), disseminated chalcopyrite (~100 μm), disseminated pyrrhotite (~100 μm) (Figure 5f). |

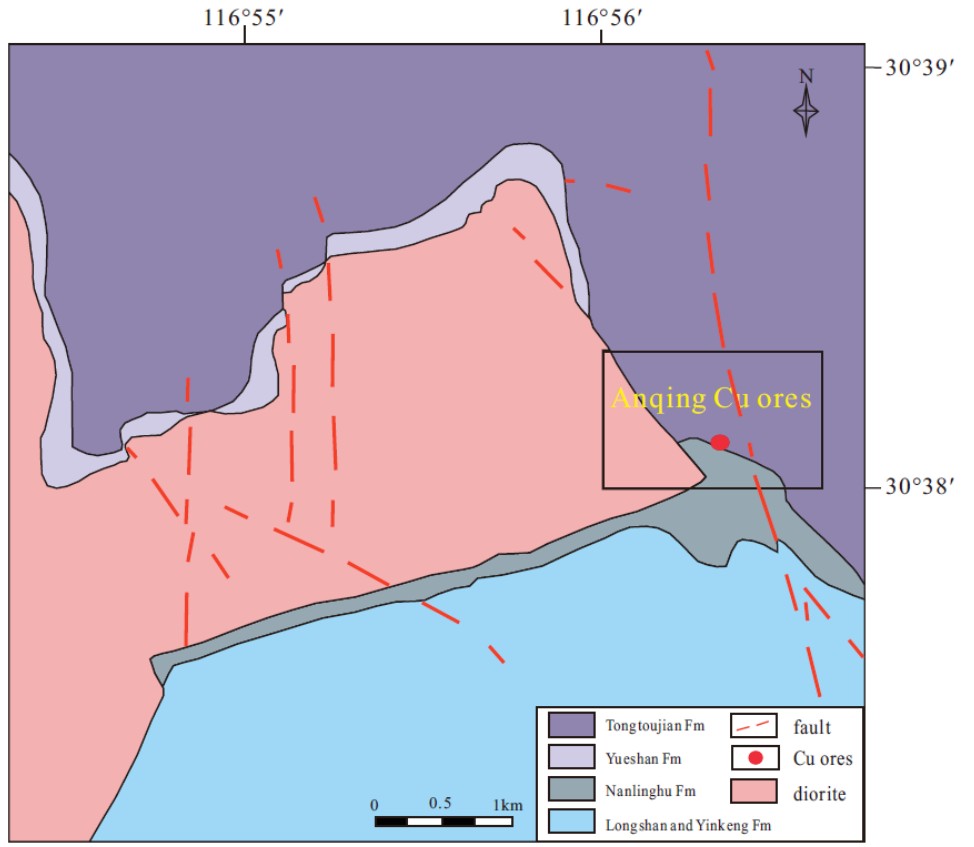

**Figure 4.** Geological map of Anqing Cu skarn deposit in eastern continental China. Fm: Formation. Ref. [61]).

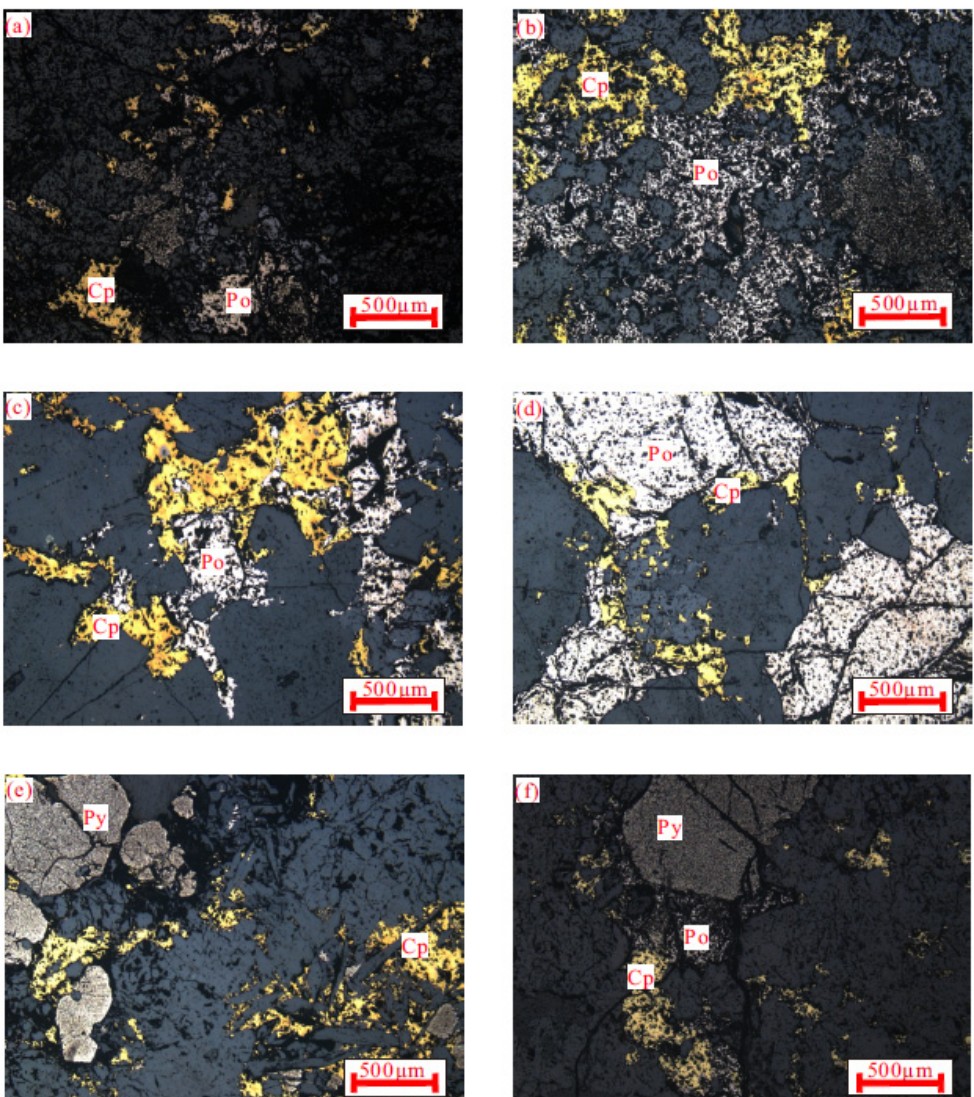

**Figure 5.** Photomicrographs of selected hydrothermal mineral assemblages from Tonglvshan and Anqing ore samples. Py = pyrite; Cp = chalcopyrite; Po = pyrrhotite. (**a**) skarn ore sample(AQ-1) with disseminated chalcopyrite and disseminated pyrrhotite.(**b**) skarn ore sample(AQ-3) with anhedral chalcopyrite, anhedral pyrrhotite and anbhedral pyrite.(**c**) skarn ore sample(AQ-5) with anhedral chalcopyrite and anhedral pyrrhotite.(**d**) skarn ore sample(AQ-7) with anhedral chalcopyrite and subhedral pyrrhotite.(**e**) skarn ore sample(AQ-8) with subhedral pyrite and disseminated chalcopyrite.(**f**) skarn ore sample(AQ-9) with subhedral pyrite, disseminated chalcopyrite and disseminated pyrrhotite.

## 3. Analytical Method

Fe isotope compositions of the ore mineral separates were analyzed in the Laboratory of Ocean Lithosphere and Mantle Dynamics, Institute of Oceanology, Chinese Academy of Sciences (IOCAS), Qingdao, China. The skarn ore samples were crushed using a corundum jaw crusher. The separation was done by using a magnetic separator to separate magnetite and pyrrhotite from non-magnetic minerals. Individual mineral phases were handpicked under a binocular microscope [36]. We have altogether obtained mineral separates of garnet, magnetite, pyrite, chalcopyrite and pyrrhotite from the massive skarn ore samples.

For Fe isotopes analysis, about 50 mg of mineral samples were digested in HNO3-HCl-HF (1 mL $H_2[(N_3O_8)Cl]$ and 0.5 mL HF) at ~190 °C for 15 h and then re-dissolved with the distilled 3 mol $L^{-1}$ $HNO_3$ for two hours until complete dissolution after evaporation. Finally, the sample were dissolved in the 1 mL 9 mol $L^{-1}$ HCl for chromatographic separa-

tion for Fe. We used a column filled with 1 mL Bio-Rad AG-MP-1 M resin (200–400 mesh) to purify the Fe element in a HCl medium following the procedures in Ref. [62]. After total purification, the eluted Fe solutions were analyzed using ICP-OES to ensure purity and full recovery. Prior to analysis, each sample was doped with GSB Ni standard (an ultrapure single elemental standard solution from the China Iron and Steel Research Institute) with Ni:Fe = 1.4:1 to monitor the instrumental mass bias during the analysis using a Nu Plasma MC-ICP-MS with wet nebulization in medium resolution (a mass resolution > 8000). Mass bias fractionation was corrected by $^{60}Ni/^{58}Ni$ similar to [63], with the $^{58}Fe$ interference on $^{58}Ni$ corrected for based on $^{56}Fe$. Each sample was analyzed five times and every two sample solutions were further bracketed with 14 ppm GSB Fe standard solution that was also doped with the GSB Ni solution with Ni:Fe ratio of 1.4:1 (a substitution of IRMM-014; $\delta^{56}Fe_{IRMM-014} = \delta^{56}Fe_{GSB} + 0.729$; $\delta^{57}Fe_{IRMM-014} = \delta^{57}Fe_{GSB} + 1.073$) [64]. Iron isotopic composition is expressed in δ-notation and normalized to IRMM-014 value: $\delta^{i}Fe(‰) = [(^{i}Fe/^{54}Fe)_{sample}/(^{i}Fe/^{54}Fe)_{IRMM-014} - 1] \times 1000$, where i refers to mass 56 or 57. The $\delta^{56}Fe$ values of the USGS standard GSP-2, BCR-2, AGV-2 and BHVO-2 were $0.13 \pm 0.04‰$, $0.07 \pm 0.04‰$, $0.09 \pm 0.04‰$, $0.11 \pm 0.02‰$, respectively, which are consistent with the literature values within error [64–67]. Instrumental and analytical details are given in [62].

## 4. Results

The Fe isotope compositions for mineral separates of the Tonglvshan Cu-Fe skarn and Anqing Cu skarn ore samples are given in Table 3. All the data are plotted on the $\delta^{57}Fe$-$\delta^{56}Fe$ mass-dependent fractionation line, demonstrating the good data quality (Figure 6).

**Table 3.** Iron isotopic composition of minerals separates from the Tonglvshan and Anqing skarn samples.

| Sample | Mineral Separates | | | | | | | | | | | |
|---|---|---|---|---|---|---|---|---|---|---|---|---|
| | Magnetite | | | | Chalcopyrite | | | | Pyrite | | | |
| | $\delta^{56}Fe$ | 2 s.d. | $\delta^{57}Fe$ | 2 s.d. | $\delta^{56}Fe$ | 2 s.d. | $\delta^{57}Fe$ | 2 s.d. | $\delta^{56}Fe$ | 2 s.d. | $\delta^{57}Fe$ | 2 s.d. |
| TLS3-1 | 0.131 | 0.036 | 0.191 | 0.068 | −0.307 | 0.033 | −0.413 | 0.076 | −0.177 | 0.041 | −0.312 | 0.051 |
| TLS3-3 | 0.018 | 0.015 | −0.002 | 0.052 | −0.317 | 0.043 | −0.495 | 0.094 | −0.063 | 0.041 | −0.086 | 0.057 |
| TLS3-5 | 0.137 | 0.016 | 0.245 | 0.035 | −0.349 | 0.047 | −0.542 | 0.040 | −0.157 | 0.018 | −0.243 | 0.071 |
| TLS3-6 | 0.075 | 0.034 | 0.105 | 0.075 | −0.296 | 0.034 | −0.454 | 0.062 | −0.062 | 0.034 | −0.121 | 0.063 |
| TLS3-9 | | | | | | | | | 0.060 | 0.049 | 0.076 | 0.086 |
| TLS3-11 | 0.196 | 0.022 | 0.258 | 0.084 | −0.381 | 0.039 | −0.499 | 0.059 | −0.156 | 0.025 | −0.214 | 0.070 |
| TLS3-12 | 0.204 | 0.029 | 0.301 | 0.056 | −0.292 | 0.054 | −0.485 | 0.076 | −0.202 | 0.027 | −0.321 | 0.070 |
| TLS4-7 | −0.113 | 0.017 | −0.119 | 0.075 | −0.837 | 0.033 | −1.209 | 0.106 | −0.538 | 0.040 | −0.790 | 0.040 |

| Sample | Mineral Separates | | | | | | | | | | | |
|---|---|---|---|---|---|---|---|---|---|---|---|---|
| | Bornite | | | | Garnet | | | | Siderite | | | |
| | $\delta^{56}Fe$ | 2 s.d. | $\delta^{57}Fe$ | 2 s.d. | $\delta^{56}Fe$ | 2 s.d. | $\delta^{57}Fe$ | 2 s.d. | $\delta^{56}Fe$ | 2 s.d. | $\delta^{57}Fe$ | 2 s.d. |
| TLS3-1 | | | | | 0.094 | 0.047 | 0.151 | 0.105 | | | | |
| TLS3-3 | | | | | −0.011 | 0.106 | −0.098 | 0.113 | | | | |
| TLS3-5 | | | | | −0.041 | 0.044 | −0.101 | 0.087 | | | | |
| TLS3-6 | | | | | 0.092 | 0.041 | 0.117 | 0.099 | | | | |
| TLS3-9 | | | | | −0.019 | 0.079 | −0.039 | 0.081 | | | | |
| TLS3-11 | | | | | −0.050 | 0.045 | −0.067 | 0.072 | | | | |
| TLS3-12 | | | | | 0.041 | 0.012 | 0.067 | 0.042 | | | | |
| TLS4-7 | −1.145 | 0.018 | −1.699 | 0.024 | −0.189 | 0.033 | −0.277 | 0.079 | −1.162 | 0.031 | −1.709 | 0.060 |

| Sample | Mineral Separates | | | | | | | | | | | |
|---|---|---|---|---|---|---|---|---|---|---|---|---|
| | Pyrrhotite | | | | Chalcopyrite | | | | Pyrite | | | |
| | $\delta^{56}Fe$ | 2 s.d. | $\delta^{57}Fe$ | 2 s.d. | $\delta^{56}Fe$ | 2 s.d. | $\delta^{57}Fe$ | 2 s.d. | $\delta^{56}Fe$ | 2 s.d. | $\delta^{57}Fe$ | 2 s.d. |
| AQ-1 | −0.399 | 0.017 | −0.557 | 0.108 | | | | | −0.123 | 0.043 | −0.200 | 0.020 |
| AQ-2 | −0.378 | 0.018 | −0.573 | 0.061 | −0.038 | 0.017 | −0.024 | 0.056 | −0.582 | 0.041 | −0.888 | 0.057 |
| AQ-3 | −0.984 | 0.015 | −1.502 | 0.030 | 0.373 | 0.049 | 0.533 | 0.043 | 0.848 | 0.067 | 1.304 | 0.107 |
| AQ-5 | −0.331 | 0.060 | −0.480 | 0.154 | 1.215 | 0.066 | 1.859 | 0.082 | | | | |
| AQ-7 | −0.251 | 0.027 | −0.395 | 0.075 | 1.308 | 0.044 | 1.901 | 0.051 | | | | |
| AQ-8 | −0.334 | 0.046 | −0.491 | 0.047 | | | | | | | | |
| AQ-9 | −1.072 | 0.057 | −1.675 | 0.125 | 0.023 | 0.026 | −0.007 | 0.066 | −0.047 | 0.032 | −0.115 | 0.069 |

Two-standard deviation (2 s.d.).

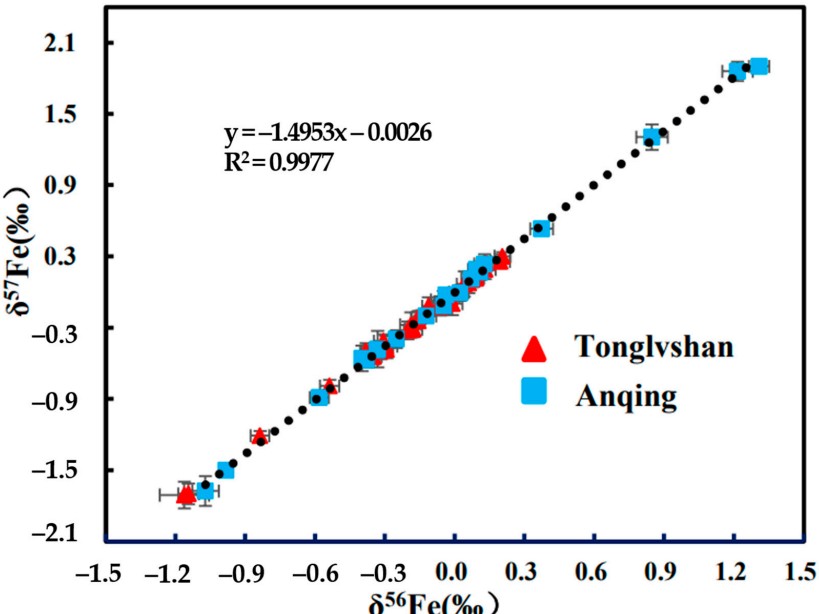

**Figure 6.** Plot of $\delta^{56}$Fe vs. $\delta^{57}$Fe of Tonglvshan and Anqing samples analyzed, defining a line that is perfectly consistent with the theoretical mass dependent fractionation.

Our Tonglvshan Cu-Fe skarn deposit samples were from two ore bodies. In samples (TLS3-1, 3-3, 3-5, 3-6, 3-11 and 3-12) from ore body III, magnetite shows a limit range of $\delta^{56}$Fe (0.018 $\pm$ 0.015 to 0.204 $\pm$ 0.029‰), which is negatively corelated with $\delta^{56}$Fe of the coexisting pyrite ($-$0.202 $\pm$ 0.027 to 0.060 $\pm$ 0.049‰) (Figures 7 and 8a). Chalcopyrite ($-$0.381 $\pm$ 0.039 to $-$0.292 $\pm$ 0.054‰) and Garnet ($-$0.050 $\pm$ 0.045 to 0.094 $\pm$ 0.047‰) have a relatively uniform $\delta^{56}$Fe signature. For sample TLS4-7 from ore body IV, we analyzed $\delta^{56}$Fe for magnetite ($-$0.113‰ $\pm$ 0.017), garnet ($-$0.189‰ $\pm$ 0.033), pyrite ($-$0.538 $\pm$ 0.040‰), chalcopyrite ($-$0.837‰ $\pm$ 0.033), bornite ($-$1.145‰ $\pm$ 0.018) and siderite ($-$1.162 $\pm$ 0.031‰) (Table 3; Figure 7). The latter two minerals had the lowest $\delta^{56}$Fe values.

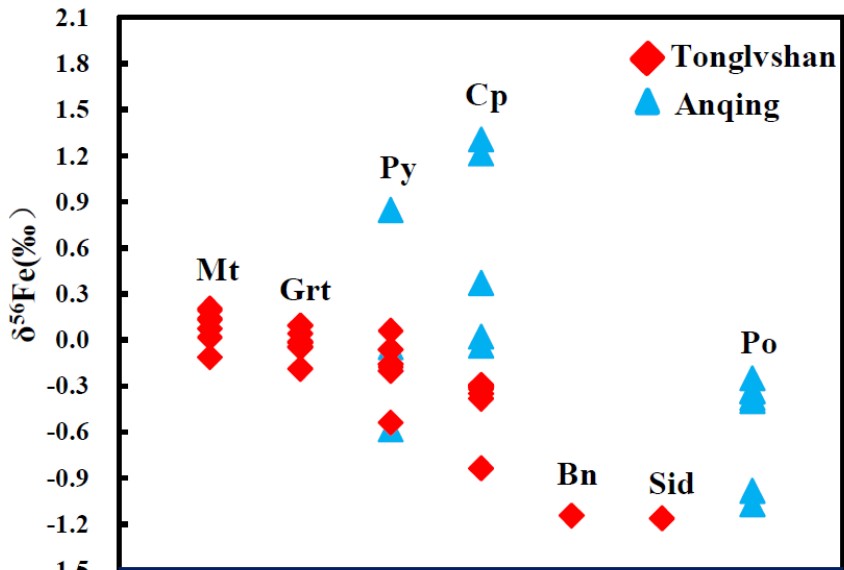

**Figure 7.** Iron isotope ratios of magmatic-hydrothermal mineral separates on samples from the Tonglvshan and Anqing ore deposits. Mt = magnetite; Py = pyrite; Cp = chalcopyrite; Po = pyrrhotite; Bn = Bornite; Grt = garnet; Sid = Siderite.

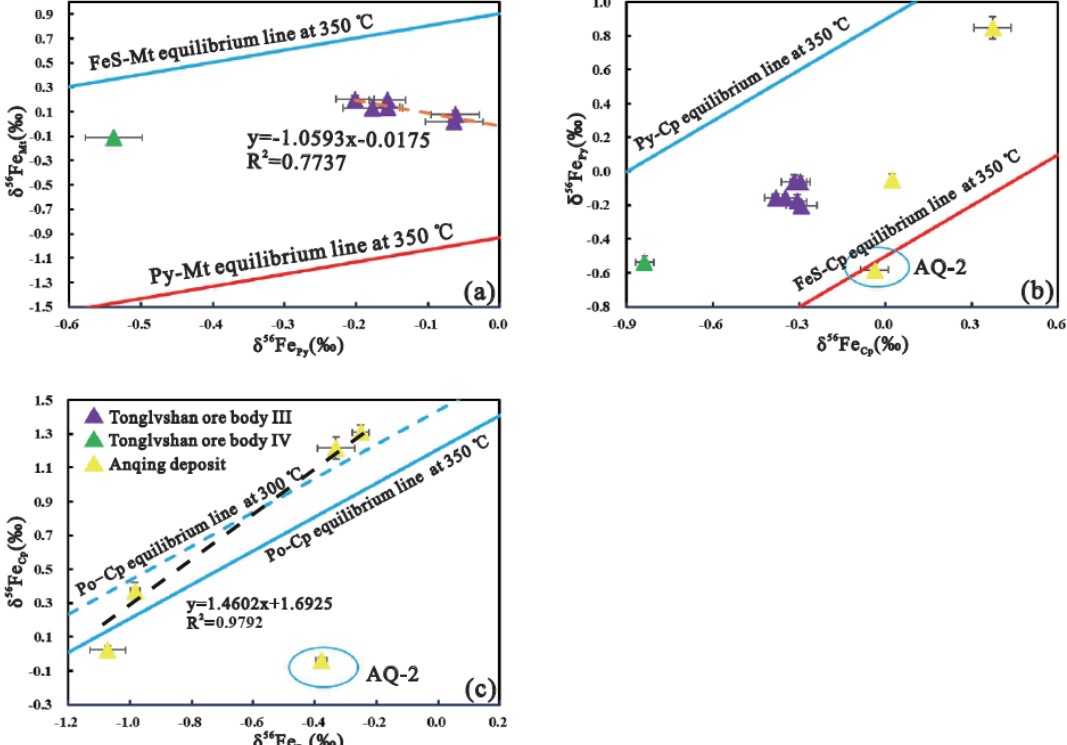

**Figure 8.** Plot of (a) $\delta^{56}Fe_{Py}$ vs. $\delta^{56}Fe_{Mt}$, (b) $\delta^{56}Fe_{Cp}$ vs. $\delta^{56}Fe_{Py}$, (c) $\delta^{56}Fe_{Po}$ vs. $\delta^{56}Fe_{Py}$ and (d) $\delta^{56}Fe_{Po}$ vs. $\delta^{56}Fe_{Cp}$. Mt = magnetite; Py = pyrite; Cp = chalcopyrite; Po = pyrrhotite. The blue and red lines stand for the equilibrium fractionation line between mineral A and B. $\Delta^{56}Fe_{A-B} = \delta^{56}Fe_A - \delta^{56}Fe_B = 10^3 ln\beta_A - 10^3 ln\beta_B$. (a) The β-factors for natural pyrite are from [42] and magnetite and troilite (FeS) are from Ref. [43]. (b) The $\Delta^{56}Fe_{Py-Cp}$ at 350 °C is from experimental studies [44,45] and the β-factors for FeS (troilite) and chalcopyrite are from [42] and [43]. (c) The $\Delta^{56}Fe_{Cp-Po}$ at 350 °C is from experimental results [45,46].

For samples from the Anqing skarn Cu deposit, we analyzed $\delta^{56}Fe$ for pyrrhotite ($-1.072‰ \pm 0.057$ to $-0.251 \pm 0.027‰$), chalcopyrite ($0.023 \pm 0.026‰$ to $1.308 \pm 0.044‰$) and pyrite ($-0.582 \pm 0.041‰$ to $0.848 \pm 0.067‰$), all displaying large variation (Table 3; Figure 7).

## 5. Discussion

### 5.1. Iron Isotope Compositions of Mineral Separates

5.1.1. Tonglvshan Cu-Fe Skarn Deposit

In the Tonglvshan Cu-Fe skarn deposit, the $\delta^{56}Fe$ values vary in the order of Mt~Grt > Py > Cp > Sid~Bn (Table 3; Figure 7). The $\delta^{56}Fe$ values for mineral separates from TLS4-7 are generally lighter than those of mineral separates from other seven samples (Table 3), which may reflect a two-stage magmatic-hydrothermal event [53]. One magmatic-hydrothermal event may be related to the samples from ore body III, whose ore-forming fluid has a heavier Fe isotope composition. The other event is possibly concerned with the TLS4-7, as the ore-forming fluid has a lighter Fe isotope composition.

The magnetite has the heaviest Fe isotopic composition, which is consistent with its high 56Fe β-factor [43]. Therefore, we considered that the magnetite may have reached Fe iso-topic equilibrium. Only one co-existing chalcopyrite and bornite were analyzed on sample (TLS4-7) to show the Fe isotope contrast between chalcopyrite and bornite. This observation ($\Delta^{56}Fe_{Cp-Bn} \approx 0.31 \pm 0.05‰$) is consistent with those in the literature [24]. The siderite has the lightest Fe ($\delta^{56}Fe = -1.16 \pm 0.03‰$), which is consistent with the prediction that ferrous carbonates preferentially incorporate the light Fe isotope [32,34].

However, according to theoretical and experimental studies [42,43,68,69], pyrite is expected to have the heaviest Fe isotopes among co-existing minerals in equilibrium

because pyrite has the highest $^{56}$Fe β-factor. There is a mainstream view to explain the 'light pyrite', whose rapid precipitation may preserve the isotopic composition of FeS precursor with light Fe, as the result of kinetic control [44,48–50]. Nevertheless, with the continuous evolution of hydrothermal fluids, equilibrium fractionation will also cause a great change in the Fe isotope composition of pyrite, which may not need to invoke kinetic fractionation (Section 5.4; see below).

5.1.2. Anqing Cu Skarn Deposit

In the Anqing Cu-Fe skarn deposit, pyrrhotite has the lightest Fe isotope composition, which is consistent with the results from the other deposits [24,32,33] and the experimental Fe isotope fractionation factor [46]. Chalcopyrite has the heaviest Fe isotope composition. Except for one chalcopyrite-pyrrhotite mineral pair, the other co-existing chalcopyrite-pyrrhotite have a good positive $^{56}$Fe correlation (Figure 8c), indicating that chalcopyrite and pyrrhotite may reach Fe isotopic equilibrium. The variation of Fe isotopic composition of pyrite is the largest, which may be explained by the Rayleigh fractionation or kinetic fractionation, if any. Detail discussions are in Section 5.4.

*5.2. The Pathway Effects for Pyrites Formation*

In nature, pyrite can be formed through multiple pathways. The detailed reactions are as follows:

$$Fe^{2+}_{(aq)} + H_2S = FeS_{(s)} + 2H^+_{(aq)} \tag{1}$$

The first precipitate is not pyrite, but the unstable mineral FeS (mackinawite) [50,70]. Then, dissolution of $FeS_s$ produces $FeS_{aq}$, which forms pyrite through the $H_2S$ pathway or polysulfide pathway [71–73]:

$$FeS_{(s)} \rightarrow FeS_{(aq)} \tag{2}$$

$H_2S$ pathway:

$$FeS_{(aq)} + H_2S = FeS_{2(s)} + H_2 \tag{3}$$

Polysulfide pathway:

$$FeS_{(aq)} + S_n{}^{2-} = FeS_{2(s)} + S_{n-1}{}^{2-} \tag{4}$$

A previous study [31] on pyrite of the Tongshankou porphyry deposit suggests that there is no obvious kinetic Fe isotopes fractionation for Reactions (3) and (4). Experimental studies show that the primary rapidly precipitated pyrite would inherit the Fe isotopic composition of the intermediate FeS phase that has a lighter Fe isotopic composition in equilibrium with the fluid [44]. However, as the reaction proceeds, the Fe isotope fractionation during pyrite formation moves towards the pyrite-fluid equilibrium with an increasing extent of the Fe isotopic exchange [31,44]. The Fe isotopic composition of the fluid is up to the extent of Fe isotopic exchange between pyrite and the fluid. The greater the extent of Fe isotopic exchange between pyrite and fluid is, the heavier the Fe isotopic composition of the pyrite and the lighter the Fe isotopic composition of the fluid will be, and vice versa. The study [31] found that the Fe isotopic composition of pyrite and chalcopyrite mineral pairs in the Tongshankou porphyry deposits are negatively correlated, which was interpreted to reflect pyrite formation through the FeS pathway in combination with S isotope data. The wide Fe isotope range of pyrite may be related to different extents of reactions during pyrite precipitation from an initial FeS-fluid equilibrium towards pyrite-fluid equilibrium, governing the Fe isotopic composition of ore-forming fluid, which in turn affects the chalcopyrite. In brief, the FeS-pathway effect is an important mechanism controlling the Fe isotope fractionation.

In the Tonglvshan skarn deposit, there is an interesting phenomenon that the Fe isotopic composition of magnetite and pyrite has a good negative correlation (Figure 8a) despite a limited Fe isotope variation. These magnetite crystals are characterized by abundant porous pits and are mostly replaced by sulfides (Figure 3), which may be caused by the dissolution–reprecipitation process (DRP) [57–59]. Hu et al. [57] suggest that DRP

may be important in the formation of hydrothermal magnetite. During such a process, the primary magnetite may exchange the Fe isotopes with the fluid whose Fe isotope composition is governed by the pyrite formation pathways [31,44]. Heavy-Fe pyrite causes the fluid to be Fe-light, which in turn causes the reprecipitation magnetite to be Fe-light. The light-Fe pyrite causes the reprecipitation magnetite to be Fe-heavy, finally resulting in the negative correlation between the magnetite and pyrite. The Fe isotope fractionation between chalcopyrite and pyrite is offset from the equilibrium line at 350 °C and lies between the FeS-chalcopyrite equilibrium line and pyrite-chalcopyrite equilibrium line at 350 °C (Figure 8b). This is best understood as the FeS-pathway effect. The extent of Fe isotope exchange between the pyrite and the fluid controls the Fe isotope composition of the fluid, further affecting the chalcopyrite.

In the Anqing skarn deposit, the Fe isotope fractionation of chalcopyrite-pyrite lying between the FeS-chalcopyrite equilibrium line and pyrite-chalcopyrite equilibrium line at 350 °C may display the role of the FeS-pathway effect (Figure 8b). Except for one chalcopyrite-pyrrhotite pair (sample AQ-2), the other co-existing chalcopyrite-pyrrhotite pairs have a good positive $^{56}$Fe correlation (Figure 8c), indicating that chalcopyrite and pyrrhotite may reach Fe isotopic equilibrium at 300 °C–350 °C.

In summary, the FeS pathway of pyrite formation in the skarn deposit may be a common mechanism causing a change in the isotopes' composition of the fluid, which in turn affects the Fe isotopic composition of the fluid and other co-existing minerals.

### 5.3. What Are the Controlling Variables That Govern the Fe Isotope Variation in Co-Exiting Phases in a Magmatic-Hydrothermal Fluid System?

With the continued cooling and crystallization of the magma, the magmatic-hydrothermal fluids are released, resulting in ore deposit formation [32,33,74,75]. During this process, Fe is transported as $Fe^{2+}$-chloride complexes [76,77]. There exist two main views about the Fe isotope fractionation during the fluid exsolution. One is the "light fluid" hypothesis, mainly confirmed by several studies [21–25,27,31], while a recent study [32] considered that the redox state actually governs the fluid Fe isotope composition. Another study [33] proposed that the oxygen and sulfur redox state of ore fluids also have an influence on isotope values of mineral assemblages because of the presence or absence of pyrrhotite (Figures 9 and 10a).

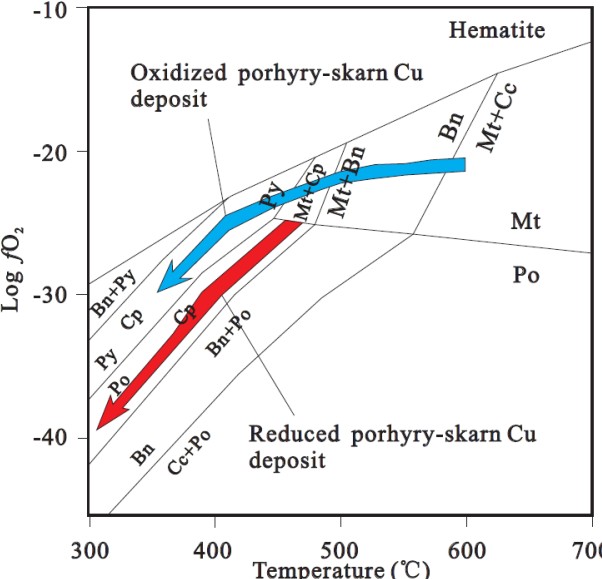

**Figure 9.** Temperature (°C) vs. oxygen fugacity (log$fO_2$) diagrams [27,33,56]. The blue path shows the mineral precipitation sequence of the oxidized porphyry-skarn Cu deposit (e.g., Tonglvshan skarn deposit) and the red path represents the path of the reduced porphyry-skarn Cu deposit (e.g., Anqing skarn deposit).

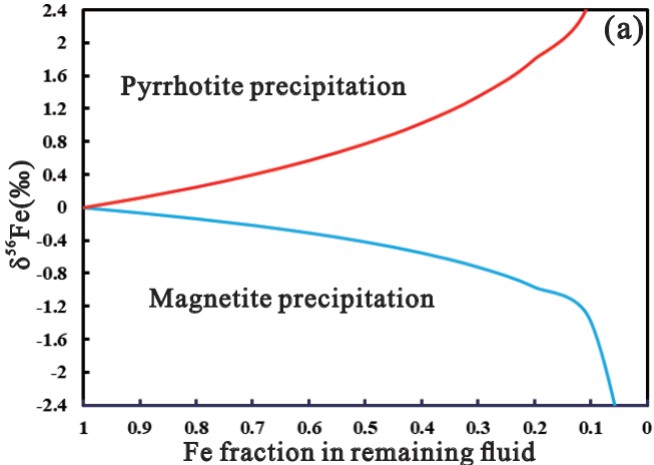 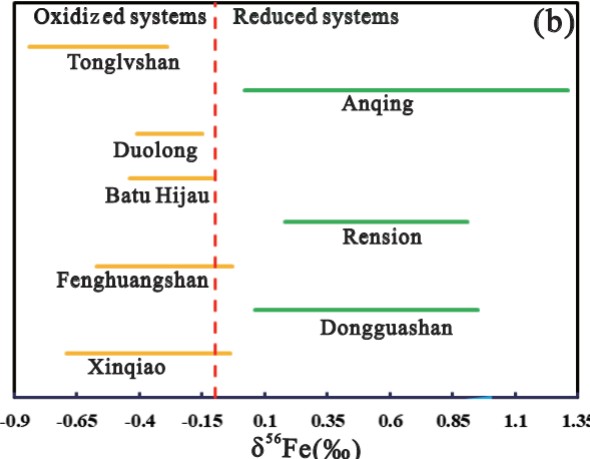

**Figure 10.** (**a**) Rayleigh fractionation modeling showing the $\delta^{56}$Fe variations of fluid controlled by the precipitation of magnetite and pyrrhotite [27], the $\Delta^{56}$Fe$_{\text{fluid-Po}}$ =1.12‰ at 350 °C is from [46] and the $\Delta^{56}$Fe$_{\text{fluid-Mt}}$ = −0.6‰ is calculated using literature data [43,69]. The initial $\delta^{56}$Fe values of fluid are assumed to be 0.0‰ and the $\delta^{56}$Fe$_{\text{fluid}}$ = $\delta^{56}$Fe$_{\text{fluid}}$ (initial) + $\Delta^{56}$Fe$_{\text{mineral-fluid}}$ × ln(F). F is the Fe mass fraction in the remaining fluid and the $\delta^{56}$Fe fluid is the Fe isotope composition of the remaining fluid. (**b**) The $\delta^{56}$Fe of the chalcopyrite from this study and several other skarns and porphyry deposits [23,24,27,32,33]. The yellow line stands for the chalcopyrite from oxidized systems and the green line represents the same for the reduced systems. The red dashed line approximates the division between the two different systems.

A recent study [27] summarizes several magmatic-hydrothermal ore deposits and proposes to use the Fe isotopic composition of chalcopyrite ($\delta^{56}$Fe $_{\text{chalcopyrite}}$~−0.1‰) as a limiting value (Figure 10b) to judge the redox conditions of deposits. Using this criterion, the Tonglvshan skarn deposit would be regarded as representing an oxidized hydrothermal system and the Anqing skarn deposit would reflect a reduced system. Combined with the phase diagram (Figure 9) and petrographic analysis (Figures 3 and 5) in the oxidized system, magnetite is the mineral that precipitates first, while in the reductive system, pyrrhotite precipitates first. Hence, magnetite from the oxidized-Tonglvshan skarn deposit is used to calculate the equilibrium fluid Fe isotope composition ($\delta^{56}$Fe$_{\text{fluid}}$ = $\delta^{56}$Fe$_{\text{mt}}$ + $\Delta^{56}$Fe$_{\text{Mt-fluid}}$, the $\Delta^{56}$Fe$_{\text{Mt-fluid}}$ = 0.6‰ at 350 °C), obtaining the heaviest $\delta^{56}$Fe ~−0.393 ± 0.029‰ [43,69], which is consistent with Ref. [41]. In the reduced-Anqing skarn deposit, we chose pyrrhotite to calculate the equilibrium fluid Fe isotope composition ($\delta^{56}$Fe$_{\text{fluid}}$ = $\delta^{56}$Fe$_{\text{Po}}$ + $\Delta^{56}$Fe$_{\text{Po-fluid}}$, the $\Delta^{56}$Fe$_{\text{Po-fluid}}$ = −1.12‰ at 350 °C) [46] and obtained the lightest $\delta^{56}$Fe ~0.045 ± 0.057‰, which is similar to the results in [41], within a rate of error. The calculated Fe isotope compositions of the mineralization fluids for both deposits are lighter than the stock intrusions associated with skarns and also porphyry mineralization [23,24,27,32,33], which is in support of the "light fluid" hypothesis.

The chalcopyrite in the oxidized-Tonglvshan skarn deposit is lighter than that from the reduced-Anqing skarn deposit, which is best understood as relating to the redox state of ore fluids because of earlier precipitated heavy Fe magnetite from the oxidized-Tonglvshan system and because of earlier precipitated light Fe pyrrhotite from the reduced-Anqing system (Table 3, Figures 7, 9 and 10) [27,32,33]. In the Tonglvshan skarn deposit, the deposited magnetite causes the fluid to become lighter, and the subsequently precipitated chalcopyrite thus has a lighter Fe isotopic composition, while in the Anqing skarn deposit, pyrrhotite incorporates light Fe isotopic composition, causing the fluid to become heavier-Fe and causes the chalcopyrite to have a heavier Fe isotopic composition (Figure 10). Moreover, the wide range of $\delta^{56}$Fe in pyrrhotite and chalcopyrite from the reduced-Anqing skarn deposit may be contributed to Rayleigh fractionation of pyrrhotite during the hydrothermal evolution (Figure 10a) [27]. The overlap of the Fe isotopic composition of pyrite in the two deposits may be related to the formation pathway of pyrite and the relative timing of the fluid evolution (Figure 8).

### 5.4. Re-Evaluating Fe Isotope Behaviors in the Skarn Mineralization

Some studies have shown that the Fe isotopic composition of pyrite in a single deposit system varies greatly. For example, pyrite in the Xinqiao Cu-Fe-Au skarn deposit has $\delta^{56}$Fe = −0.83‰ to 0.46‰; pyrite in the Dongguashan Cu-Au skarn-porphyry deposit has $\delta^{56}$Fe = −0.21‰ to 1.58‰; and pyrite in the Fenghuangshan Cu-Fe-Au skarn deposit has $\delta^{56}$Fe = −0.55‰ to 1.46‰. [22,23]. He et al. [31] suggest that such large variation may reflect the involvement of low-temperature sedimentary materials. Wei et al. [36] divides pyrites of the Damiao titanomagnetite ore deposit into two types, magmatic pyrite ($\delta^{56}$Fe: −0.638‰ to −0.056‰) and hydrothermal pyrite ($\delta^{56}$Fe: 0.435‰ to 0.662‰).

In Section 3 above, we discussed the redox state of the ore fluid that governs the Fe isotopes composition of precipitated minerals [33]. In the Tonglvshan deposit, precipitation of magnetite with heavy Fe will cause the fluid to become lighter, which will impart lighter Fe in subsequently precipitated chalcopyrite from this fluid. In the Anqing deposit, precipitation of pyrrhotite with light Fe can cause the fluid to become heavier (Figure 10) [27], which is expected to have great influence on the latter precipitated chalcopyrite. This is important. If we assume an initial $\delta^{56}$Fe value of 0‰ for fluid, magnetite precipitation can deplete the heavier Fe, resulting in light Fe in the liquid, readily reaching a value of $\delta^{56}$Fe = ~−0.3‰ (Figure 10). Likewise, precipitation of pyrrhotite can deplete the light Fe, resulting in heavy Fe in the liquid, readily reaching a value of $\delta^{56}$Fe = ~0.3‰ (Figure 10).

As mentioned in Section 1, we suggest equilibrium fractionation will also cause a great change in the Fe isotopic composition of pyrite (Figure 11a). With the FeS pathway mechanisms, i.e., the transient FeS-fluid equilibrium followed by pyrite-fluid equilibrium, we can readily explain the large Fe isotope compositional variation of pyrites in skarn deposits (Figure 11). The $\delta^{56}$Fe for pyrites can be simply contributed to the Rayleigh fractionation model by the combination of pyrite-fluid and FeS-fluid equilibrium fractionations (Figure 11). Our model shows that the pyrite-fluid equilibrium fractionations result in the light Fe isotopes boundary of the pyrite (Figure 11a,b), while the heavy Fe isotopes boundary of the pyrite is caused by the FeS-fluid equilibrium fractionations (Figure 11c,d). With all the observations and discussion above considered, we advocate that that the FeS-pathway effects on pyrite formation may be a common mechanism controlling the fluid Fe isotope composition in both oxidized (i.e., Tonglvshan) and reduced (i.e., Anqing) skarn ore-forming systems.

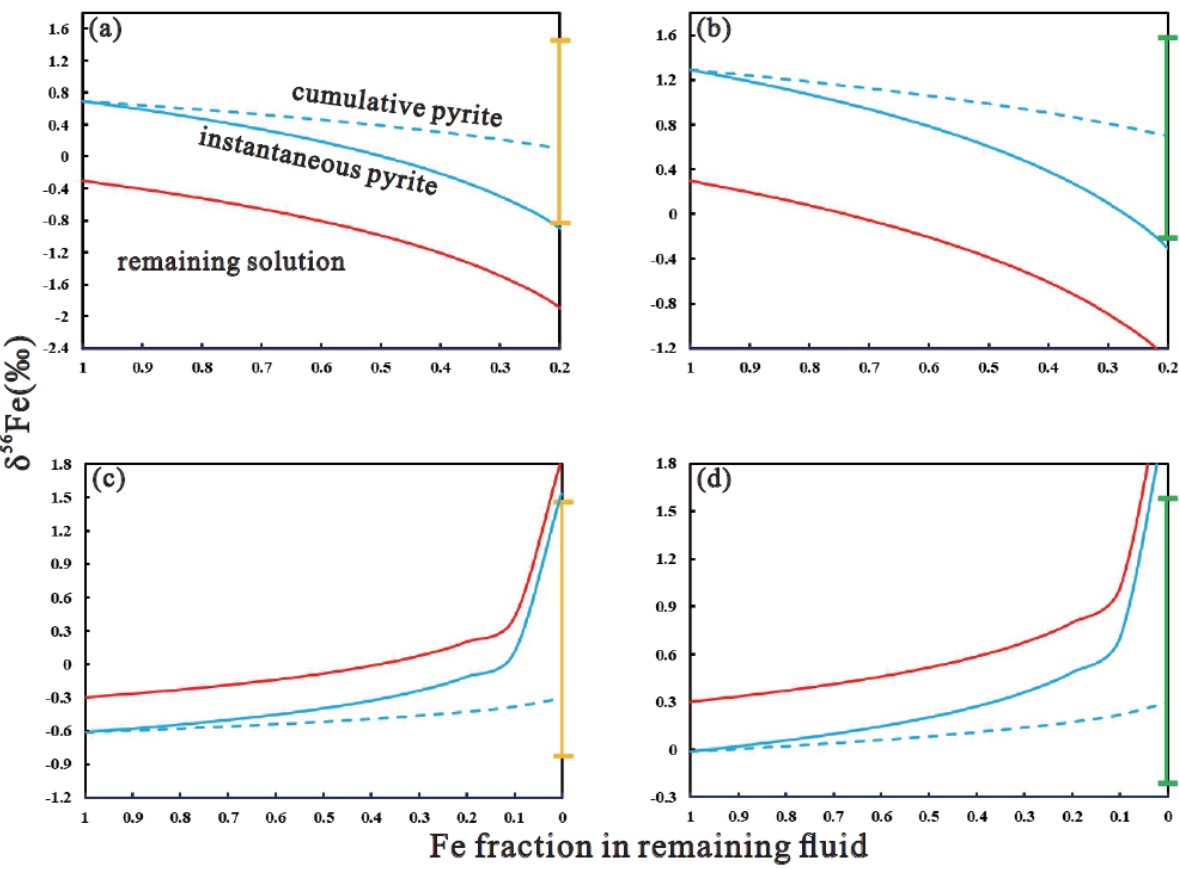

**Figure 11.** Rayleigh fractionation modeling showing that $\delta^{56}$Fe (‰) variations in the residual fluid(red solid line), instantaneous(blue solid line ) and cumulative pyrite(blue dashed line) during pyrite precipitation. Instantaneous pyrite represents single pyrite formed at every instant and cumulative pyrite means bulk pyrite at every instant. The initial fluid Fe isotope composition for the oxidized systems is set to be −0.3‰ (**a,c**) and for the reduced system is + 0.3‰ (**b,d**). (**a**) Equilibrium fractionation between pyrite and fluid in the oxidized system; (**b**) Equilibrium fractionation between pyrite and fluid in the reduced system; (**c**) Equilibrium fractionation between FeS and fluid in the oxidized system and (**d**) Equilibrium fractionation between FeS and fluid in the reduced system. Equilibrium fractionation factors at 350 °C between pyrite and fluid is 0.99‰ from [44], while between the FeS and fluid is −0.31‰, calculated from [43,69]. The Rayleigh fractionation equations are as follows: $\delta^{56}$Fe$_{fluid}$ = $\delta^{56}$Fe$_{fluid}$ (initial) + $\Delta^{56}$Fe$_{mineral\text{-}fluid}$ × ln(F); $\delta^{56}$Fe$_{mineral}$ (instantaneous) = $\delta^{56}$Fe$_{fluid}$ + $\Delta^{56}$Fe$_{mineral\text{-}fluid}$; $\delta^{56}$Fe$_{mineral}$ (cumulative) = [$\delta^{56}$Fe$_{fluid}$ (initial) − (1 − F) × $\delta^{56}$Fe$_{fluid}$]/F; F is the Fe mass fraction in the remaining fluid. The yellow line stands for the Fe isotope compositional range of pyrite from the oxidized system and the green line represents the same for the reduced system [22,23].

## 6. Conclusions

This work, based on Fe isotope data on skarn ore mineral separates and petrography, led to the following important conclusions.

1.  We used magnetite from the oxidized-Tonglvshan skarn deposit and pyrrhotite from the reduced-Anqing skarn deposit to calculate the equilibrium Fe isotope compositions of the fluids, respectively. The calculated heaviest Fe isotope of the equilibrium fluid for the oxidized-Tonglvshan skarn deposit and the lightest Fe isotope of the equilibrium fluid for the reduced-Anqing skarn deposits are both lighter than the stock intrusions associated with skarn and also porphyry mineralization, supporting the "light fluid" hypothesis for granitoid magmatic fluids. Moreover, the chalcopyrite Fe isotope in the oxidized-Tonglvshan skarn deposit is lighter than that from the reduced-Anqing skarn deposit, which is controlled by the prior precipitation of magnetite (heavy Fe) from the ore fluid in the oxidized-Tonglvshan systems and the prior precipitation of pyrrhotite (light Fe) from the ore fluid in the reduced-Anqing system.

2.   The $\delta^{56}$Fe for pyrite shows an inverse correlation with $\delta^{56}$Fe for the magnetite in Tonglvshan ore samples. In both deposits, the Fe isotope fractionation between chalcopyrite and pyrite is offset from the equilibrium line at 350 °C and lies between the FeS-chalcopyrite equilibrium line and pyrite-chalcopyrite equilibrium line at 350 °C. These observations are best understood as the FeS pathway towards pyrite formation. That is, the initial FeS-fluid equilibrium Fe isotope fractionation is a critical step for continued pyrite-fluid equilibrium fractionation via increased extent of Fe isotopic exchange. With data from other ore deposits considered altogether, we advocate that the pathway effect on pyrite formation in the skarn deposit mineralization is important in controlling the Fe isotope fractionation.

**Author Contributions:** Conceptualization, Y.N.; writing, S.X., revision, Y.N. and S.X.; additional re-writing, reviewing and editing, S.X., Y.N., Y.C., B.X., P.W. and Y.S.; analysis with assistance by S.X., H.G., X.W. and M.D.; funding acquisition, Y.N. All authors have read and agreed to the published version of the manuscript.

**Funding:** Financial support for this research was provided by the National Natural Science Foundation (NSFC, 41630968, 91958215), and NSFC-Shandong Joint Fund for Marine Science Research Centers (U1606401) as well as the 111 project (B18048).

**Data Availability Statement:** The data presented in this study are available in the text.

**Acknowledgments:** We thank Pengyuan Guo, Shuo Chen and Pu Sun for discussion and help with lab work. We thank Yi Liu, Pengyuan Guo, Shuo Chen, Fangyu Shen, Junjie Zhan, Yang Tian and the company of Daye Nonferrous Metals for collecting the samples during the 2018 annual field work. We also thank Leo Du for handling our manuscript and three journal reviewers for constructive comments.

**Conflicts of Interest:** The authors declare no conflict of interest.

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
