# Peer review of "Iron Isotope Fractionation during Skarn Cu-Fe Mineralization"

_minerals, doi:10.3390/min11050444_

Round 1

Reviewer 1 Report

Review: The manuscript compares reduced and oxidized magmatic-hydrothermal systems via Fe isotope systematics of Fe-sulfides and magnetite. The authors argue that the FeS pathway is the predominant mechanism dictating the composition of pyrite in the compared samples. The Fe isotope analyses are great. However, with regards to the interpretation of the text, I would appreciate if the authors took into consideration some of the overlying assumptions behind equilibrium between magnetite-fluid. Especially, how does the disequilibrium of pyrite in the system have an effect on the Fe isotope composition of the fluid? Pyrite is a major sulfide mineral in the system and the disequilibrium produced during its formation may also be associated with the co-existing fluid. I recommend publication of this manuscript but with minor updates to the manuscript. I apologize to the authors for my delay in review.

Line 44-45: I would probably expand this sentence to account chemical mass transfer mechanisms that isn’t just diffusion.

Line 53-54: “(1) Fe isotopic 53 fractionation during fluid exsolution [23-25,32,33]; (2) between mineral Fe isotopic fractionation [31].” I would recommend re-phrasing this sentence to be clearer on the uncertainties associated with Fe isotope fractionation, in particular, to magmatic-hydrothermal environments.

Figure 6: I recommend changing the legend “text” of “blue” and “red” to the actual symbols and their appropriate color used to designate the deposit. Make these particular symbols a bit larger than the data symbols.

Line 234: change “isotopes” to “isotope”

Line 249 – 254: The explanation for the negative correlation between magnetite and pyrite is not clear. I would try to re-write/illustrate better the process creating this correlation.

Line 255 – 258: I would combine these sentences into the following paragraph about chalcopyrite, be sure to mention that the chalcopyrite is from a different deposit show demonstrates similar characteristics in the variability of the fractionation.

Line 267: After the final sentence, I would suggest adding another sentence that describes briefly on how the fluid Fe isotope composition would respond to the FeS – pathway effect.

Lines 301 and 303: change “isotopes” to “isotope”. Also for this section of the paragraph, it would be helpful to justify why you chose magnetite and pyrrhotite equilibrium fractionation estimates to predict the fluid d56Fe composition? The data on the other plots broadly demonstrate disequilibrium for most minerals.

Line 332: change “isotopes” to “isotope”

Lines 332 – 334: There is currently no experimental research that provide Fe isotope fractionation constraints that suggest magnetite will precipitate at isotopic equilibrium. Experimental research by Syverson et al., 2017 “Serpentinization of olivine at 300 °C and 500 bars: An experimental study examining the role of silica on the reaction path and oxidation state of iron” in Chemical Geology, demonstrates that the dissolution of silicate minerals will produce a light Fe isotopic signature in the fluid and the enrichment in heavy isotopes in the fluid phase may occur due to kinetic effects that are associated with the during rapid precipitation of magnetite (which would be enriched in the light isotopes in this case, non-equilibrium).

Author Response

Reply to the Reviewer #1

The manuscript compares reduced and oxidized magmatic-hydrothermal systems via Fe isotope systematics of Fe-sulfides and magnetite. The authors argue that the FeS pathway is the predominant mechanism dictating the composition of pyrite in the compared samples. The Fe isotope analyses are great.

However, with regards to the interpretation of the text, I would appreciate if the authors took into consideration some of the overlying assumptions behind equilibrium between magnetite-fluid. Especially, how does the disequilibrium of pyrite in the system have an effect on the Fe isotope composition of the fluid? Pyrite is a major sulfide mineral in the system and the disequilibrium produced during its formation may also be associated with the co-existing fluid.

I recommend publication of this manuscript but with minor updates to the manuscript. I apologize to the authors for my delay in review.

Reply:

Thanks you for the constructive comments and suggestions, which help us to improve our manuscript.

In our Fe isotope analyses, the magnetite from Tonglvshan has the heaviest Fe isotope composition, which is consistent with its high 56Fe β-factor. Therefore, we consider that the magnetite may have reached Fe isotopic equilibrium. Some other studies of porphyry-skarn deposits indicate that magnetite may reach Fe isotopic equilibrium during mineralization [23,24,27]. We have further illustrated this in Section 5.1.1.

In the text, we advocate that the pathway effect on pyrite formation in the skarn deposit mineralization is important in controlling the Fe isotope fractionation. That is, Fe isotope fractionation during pyrite formation depends on a path, from the initial FeS-fluid equilibrium towards the pyrite-fluid equilibrium due to the increasing extent of Fe isotopic exchange with fluids. So the Fe isotope composition of the fluid is up to the extent of Fe isotopic exchange between pyrite and fluid. The higher the extent of Fe isotope exchange between pyrite and the fluid, the heavier the Fe isotopic composition of pyrite and the lighter the Fe isotopic composition of the fluid will be, and vice versa.

Reply to the main critical comments as follows:

1.Line 44-45: I would probably expand this sentence to account chemical mass transfer mechanisms that isn’t just diffusion.

Reply:

Thanks for your suggestion.

However, refs.[16,17] mainly talk about the isotope fractionation during diffusion process and we think our statement is proper.  

2.Line 53-54: “(1) Fe isotopic fractionation during fluid exsolution [23-25,32,33]; (2) between mineral Fe isotopic fractionation [31].” I would recommend re-phrasing this sentence to be clearer on the uncertainties associated with Fe isotope fractionation, in particular, to magmatic-hydrothermal environments.

Reply:

Thanks for your suggestion.  

What we mean by that is, (1) iron isotopic fractionation during fluid exsolutions, and (2) iron isotopic fractionation between minerals and fluids, and between minerals during fluid evolution.

We have clarified this in revision as follows:

(1) Fe isotope fractionation during fluid exsolution[23-25,32,33];

(2) Fe isotope behavior in fluid-mineral systems (e.g., mineral-fluid fractionation and between-mineral fractionation)

3.Figure 6: I recommend changing the legend “text” of “blue” and “red” to the actual symbols and their appropriate color used to designate the deposit. Make these particular symbols a bit larger than the data symbols.

Reply:

Thanks for your suggestion.

We have revised.

4.Line 234: change “isotopes” to “isotope”.

Reply:

Okay.

5.Line 249 – 254: The explanation for the negative correlation between magnetite and pyrite is not clear. I would try to re-write/illustrate better the process creating this correlation.

Reply:

Thanks for your suggestion.

We have revised accordingly as follows.

During such process, the primary magnetite may exchange the Fe isotopes with the fluid whose Fe isotope composition is governed by the pyrite formation pathways [31,44]. Heavy-Fe pyrite causes the fluid to be Fe-lighter, which in turn causes the reprecipitation magnetite to be Fe-lighter, and light-Fe pyrite contributes the reprecipitation magnetite to be Fe-heavier, finally resulting in the negative correlation between the magnetite and pyrite.

6.Line 255 – 258: I would combine these sentences into the following paragraph about chalcopyrite, be sure to mention that the chalcopyrite is from a different deposit show demonstrates similar characteristics in the variability of the fractionation.

Reply:

Thanks for your suggestion.

Here, we only want to address the FeS-pathway effect of pyrite formation. The following paragraph about chalcopyrite is mainly to compare the iron isotopic behavior in oxidizing and reducing deposits.

7.Line 267: After the final sentence, I would suggest adding another sentence that describes briefly on how the fluid Fe isotope composition would respond to the FeS – pathway effect.

Reply:

Thanks for your good suggestions! We have added another sentence that describes briefly on how the fluid Fe isotope composition would respond to the FeS – pathway effect after “However, as the reaction proceeds, the Fe isotope fractionation during pyrite formation moves towards the pyrite-fluid equilibrium with increasing extent of Fe isotopic exchange [31,44]”.

The Fe isotopic composition of the fluid is up to the extent of Fe isotopic exchange between pyrite and the fluid. The greater the extent of Fe isotopic exchange between pyrite and fluid is, the heavier the Fe isotopic composition of the pyrite and the lighter the Fe isotopic composition of the fluid will be, and vice versa.

8.Lines 301 and 303: change “isotopes” to “isotope”. Also for this section of the paragraph, it would be helpful to justify why you chose magnetite and pyrrhotite equilibrium fractionation estimates to predict the fluid d56Fe composition? The data on the other plots broadly demonstrate disequilibrium for most minerals.

Reply:

Thanks for your questions and suggestion!

We add the following sentences in the 5.3

Combined with the phase diagram (Fig.9) and petrographic analysis (Fig.3, Fig 5), in the oxidized system, magnetite is the mineral that precipitates first, while in the reductive system, pyrrhotite precipitates firstly.

Therefore, the iron isotopic composition of these two minerals is used to estimate the iron isotopic composition of the initial fluid.

In Fig. 8 a and b, the Fe isotope fractionations between co-existing minerals do not fall on the equilibrium fractionation line. Our interpretation is that the FeS-pathway of pyrite formation causes a change in the Fe isotope composition of the fluid, which in turn affects the Fe isotope composition of other co-existing minerals. The Fe isotope fractionation of magnetite-fluid, pyrrhotite-fluid and chalcopyrite-fluid can readily arrive at equilibrium in the hydrothermal systems as shown by others [23,24,27,32,33,44].

9.Line 332: change “isotopes” to “isotope”

Reply:

Okay.

10.Lines 332 – 334: There is currently no experimental research that provide Fe isotope fractionation constraints that suggest magnetite will precipitate at isotopic equilibrium. Experimental research by Syverson et al., 2017 “Serpentinization of olivine at 300 °C and 500 bars: An experimental study examining the role of silica on the reaction path and oxidation state of iron” in Chemical Geology, demonstrates that the dissolution of silicate minerals will produce a light Fe isotopic signature in the fluid and the enrichment in heavy isotopes in the fluid phase may occur due to kinetic effects that are associated with the during rapid precipitation of magnetite (which would be enriched in the light isotopes in this case, non-equilibrium).

Reply:

This is a good article and your proposal is useful constructive.

In this paper, with the reaction progress, an increase in pH causes dissolved Fe to become less soluble while Fe(OH)+ complex becomes the dominant species, which may become the intermediates in the formation of magnetite. Thus, the magnetite would be enriched in the light isotopes.

If so, the prior precipitation of magnetite will cause the ore-forming fluid to become Fe-heavier, and the resulting chalcopyrite should have a heavy iron isotopic composition, but this is contrary to the observations of the Tonglvshan deposit and most other deposits.

In addition, the properties of ore-forming fluids are quite different from those in the process of serpentinization, and the main phase of iron in ore-forming fluids exists in the form of chloride.

Furthermore, other studies show that the magnetite in the porphyry-skarn deposit generally has heavy iron isotopic composition, suggesting that the magnetite may reach iron isotopic equilibrium [23,24,27].

The serpentinization process is complex and if we want to fully understand the process of Fe isotope fractionation, it is necessary to develop in-situ testing techniques, which is the next step for our laboratory.

Reviewer 2 Report

The manuscript presents Fe isotope data of Fe-bearing minerals in Tonglvshan and Angquing skarn deposits to investigate factors affecting isotopic variations in the ore deposit systems. They are particularly focused on Fe isotope compositions of magnetite, pyrrhotite, pyrite, and chalcopyrite relating to their petrogenesis. They mostly attribute their Fe isotope behaviors to exsolving fluids having isotopically lighter Fe and resulting Fe isotope fractionation during mineral precipitations. On the other hand, they conclude that the contrasting behavior of chalcopyrite in the two deposits is governed by the change in the isotope ratios of mineralizing fluids affected by the precipitation of magnetite and pyrrhotite, respectively. Overall, the manuscript is well written. The data are nicely presented and discussion is sound. Therefore, I recommend publishing the manuscript from Minerals after minor revision.

There are a few discussion points that I suggest adding to the manuscript:

  • A previous study (He et al., 2029) analyzed Fe isotopes of pyrite in the Porphyry system of Tonglvshan deposit. Fe isotopic ratios of the pyrite do not likely inherit the kinetic isotope effect during precipitation of precursor (i.e., FeS) as opposed to the current study on the skarn system. Although the authors mention the difference, I would like to see more discussion why they show the different isotopic behaviors even if it might be speculative.

  • The authors compile Fe isotope ratios of chalcopyrite between oxidized and reduced systems in porphyry and skarn ore deposits from other regions (Fig. 10b) and argue that the trend observed in their study is generally applicable. However, I am suspicious that Fe isotope ratios of chalcopyrite in the oxidized systems are affected by magnetite precipitation in all the deposits. Is the suggestion also supported by petrographic observation in the other ore deposits?

Other minor points:

Title: The current title sounds too general to me.

Figures: The order of figure numbers does not accord with the order for the citation in the main text. They should be consistent.

Figure 2: What do the element’s names (e.g., Cu, Fe, C-Fe, Mo) indicate? Mineralization zone? I don’t see any place for Mo (pink) in the map.

Figures 3 and 4: The photomicrographs do not appear to be taken appropriately (e.g., too o dark pyrite, chalcopyrite). Are they polished enough? I suggest retaking the pictures.

Figure 8b Py-Cp equilibrium, not Py-Mt?

Author Response

Reply to the Reviewer #2

The manuscript presents Fe isotope data of Fe-bearing minerals in Tonglvshan and Angquing skarn deposits to investigate factors affecting isotopic variations in the ore deposit systems. They are particularly focused on Fe isotope compositions of magnetite, pyrrhotite, pyrite, and chalcopyrite relating to their petrogenesis.

They mostly attribute their Fe isotope behaviors to exsolving fluids having isotopically lighter Fe and resulting Fe isotope fractionation during mineral precipitations. On the other hand, they conclude that the contrasting behavior of chalcopyrite in the two deposits is governed by the change in the isotope ratios of mineralizing fluids affected by the precipitation of magnetite and pyrrhotite, respectively.

Overall, the manuscript is well written. The data are nicely presented and discussion is sound. Therefore, I recommend publishing the manuscript from Minerals after minor revision.

Reply:Thank you for your affirmation on this work.

Reply to the main critical comments as follows:

1.A previous study (He et al., 2020) analyzed Fe isotopes of pyrite in the Porphyry system of Tonglvshan deposit. Fe isotopic ratios of the pyrite do not likely inherit the kinetic isotope effect during precipitation of precursor (i.e., FeS) as opposed to the current study on the skarn system. Although the authors mention the difference, I would like to see more discussion why they show the different isotopic behaviors even if it might be speculative.

Reply:

Thanks for your suggestion.

He et al.,2020 analyzed Fe isotopes of pyrite in the Porphyry system of Tongshankou deposit, not the Tonglvshan deposit. The wide range of Fe isotopic ratios of the pyrite has been interpreted to reflect the FeS-pathway effect, which is related to different extent of reaction during pyrite precipitation from initial FeS-fluid equilibrium towards pyrite-fluid equilibrium.

In the Tonglvshan skarn deposit, we also agree that the pyrite’s Fe isotope composition is affected by the FeS-pathway effect and we advocate that the FeS-pathway of pyrite formation in the skarn deposit may be a common mechanism causing the Fe isotope fractionation.

2.The authors compile Fe isotope ratios of chalcopyrite between oxidized and reduced systems in porphyry and skarn ore deposits from other regions (Fig. 10b) and argue that the trend observed in their study is generally applicable. However, I am suspicious that Fe isotope ratios of chalcopyrite in the oxidized systems are affected by magnetite precipitation in all the deposits. Is the suggestion also supported by petrographic observation in the other ore deposits?

Reply:

Thanks for your question.

As far as I know, the Fe isotopic compositions of chalcopyrite in the oxidation system in most porphyry-skarn deposits are influenced by magnetite precipitation, which is best understood as the result of the prior precipitation of magnetite (heavy Fe) from the ore fluid in the oxidized systems.

From the Fig 9, the phase diagram shows that the sulfides in the porphyry-skarn systems may form later than the magnetite. The prior precipitation of magnetite must have an influence on the Fe isotope compositions of the fluid which will cause the variation of sulfides Fe isotope composition.

All petrographic observations of our oxidized deposit examples in our article (Fig 10b) show that the chalcopyrite formed later than the magnetite, which may support the conclusion that the prior precipitation of magnetite will cause ‘lighter chalcopyrite’.

3.Title: The current title sounds too general to me.

Reply:

Thanks for this suggestion. 

However, we think it is the most concise and informative title.

4.Figures: The order of figure numbers does not accord with the order for the citation in the main text. They should be consistent.

Reply:

Thanks for this suggestion and this has been resolved.

5.Figure 2: What do the element’s names (e.g., Cu, Fe, C-Fe, Mo) indicate? Mineralization zone? I don’t see any place for Mo (pink) in the map.

Reply:

Thanks for your question.

They stand for different skarn ore bodies and these have been clarified in revision.

6.Figures 3 and 4: The photomicrographs do not appear to be taken appropriately (e.g., too o dark pyrite, chalcopyrite). Are they polished enough? I suggest retaking the pictures.

Reply:

Thanks for your suggestions.

Indeed, they are not polished enough. We have tried to retake the photos, but the pyrite and chalcopyrite still look dark, so we stick to the original photos. Sorry about this, but they are well labeled.

7.Figure 8b Py-Cp equilibrium, not Py-Mt?

Reply:

Thanks for pointing this mistake!

We have corrected.

Reviewer 3 Report

Please see the attachment for the comments. 

Author Response

Reply to the Reviewer #3

The article entitled “Iron Isotope fractionation during skarn Cu-Fe mineralization” deals with an quite interesting topic; providing evidences for Fe-isotopic variations in different hydrothermal ore-bodies due to changes in redox conditions of magmatic sources. Thus based on Fe-isotopic ratios in selective minerals from these ores, the related evolutionary processes has been discussed.

The analytical data has supporting the scientific conclusion presented in this study. However, the overall presentations has some lacking’s; particularly in the discussion many statements need to be presented in more elaborated manner. This would provide more clarity to the draft and might help the readers to conceptualise the scientific knowledge, authors want to present in their manuscript

Reply:

Thanks for your helpful comments we have clarified throughout.

The language used in the present draft is fine, but still it has scope for modifications/improvements. There are many cases, where authors should avoid lengthy statements and go for simple smaller sentences. Some sentences need corrections also.

Reply:

Thanks for your suggestion. We have clarified test throughout.

The volume of abstract has to be reduced.

Reply:

Okay. We have simplified the abstract.

Moreover, in my opinion, some more technical issues (see the specific comments below) also have to be attended carefully during revision. And thus considering all these, I would like to mention that the present draft will only be publishable after considerable revisions.

Reply:

Thanks you! We have revised the manuscript wherever needed throughout the text.

Reply to the main critical comments as follows:

1.Ln 53-55: “…..two important problems remains concerning magmatic-hydrothermal deposits…..” In this line authors mentioned about major problems i.e., (1) Fe isotopic fractionation during fluid exsolution and (2) between mineral Fe isotopic fractionation. --Instead of such normal statements, authors should raise two simple scientific questions, which they are going to deal with in their study.

Reply:

Thanks for your suggestion.

However, our study mainly focuses on two major problems.

For question (1), our results support the “light fluid” hypothesis for granitoid magmatic fluids during the fluid exsolution.

For question (2), our study focuses on co-existing minerals with the aim of helping understand the Fe isotopic fractionation between minerals.

According to your suggestion and another reviewer, we change these statements to 

(1) Fe isotope fractionation during fluid exsolution[23-25,32,33];(2) Fe isotope behavior in fluid-mineral systems (e.g., mineral-fluid fractionation and between-mineral fractionation).

2.Ln 73-77: Too long statement. Better to split it into multiple sentences.

Reply:

Thanks for your suggestion

We have revised accordingly!

3.Ln 79-82: Sentence is not very clear. Go for simpler sentences.

Reply:

Thanks for your suggestion.

However, the sentence well serves the purpose.

The Middle-Lower-Yangtze River (MLYR) metallogenic belt (Fig. 1) is a famous Cu-Fe-Au porphyry-skarn province in China, from east to west, consisting of Ningzhen, Ningwu, Luzong, Tongling, Anqing-Guichi, Jiurui and Edong ore districts, all of which are associated with late Mesozoic granitoid magmatism [51,52].

4.Ln 89: The Figure 2 doesn’t have any latitude. Need to be corrected.

Reply:

Thanks for pointing out this defect.

Have done in revision.

5.Ln 97: “…….economic metals of 1.08t Cu, 60t Fe, 70t Au (0.38 g/t Au), and 508t Ag…..”. I am not very sure about these metal contents. But the availability of 70ton gold against only 1ton of total copper or 60ton iron in same volume of ore deposit is bit surprising. Authors should be careful about these numbers.

Reply:

Sorry for the errors, which have been corrected for in revision as follows,

Previous studies report that this deposit contains economic metals of 1.08Mt Cu (1.78% Cu), 60Mt Fe (41%Fe), 70t Au (0.38 g/t Au), and 508t Ag.

6.Ln 98: “……..U-Th-Pb method, shows two independent…….” This statement could be changed as “……U-Th-Pb method, shows the evidence of two independent……”.

Reply:

Thanks for your suggestions and we have clarified this in revision as follows.

A recent dating study [53], using Laser ablation ICP-MS titanite U-Th-Pb method, shows evidence of two independent hydrothermal events (~136 Ma and 121Ma).

7.Ln 100-101: “Detailed fluid inclusion work suggested magnetite deposition at 405-567℃ and quartz-sulfide mineralization at 240-350℃…” The sentence is not proper.

Reply:

I am sorry why this is is not proper, but I have made a small modification.

Detailed fluid inclusion work suggested the magnetite deposition temperature of 405-567℃ and quartz-sulfide mineralization temperature of 240-350℃[55].

8.Ln 108-111: “The magnetite is mostly replaced by sulfides, suggesting ………the paragenetic sequence may be as follows: magnetite → pyrite / chalcopyrite”…. How authors confirm such paragenetic shifts during mineral developments?? If it is based on microscopic observations, authors should provide necessary description.

Reply:

Thanks for your questions.

First, from the phase diagram in Fig. 9, we can get the sequence of mineral precipitation of the two deposits.

Second, petrographic observations show that the contact relationships between minerals indicate the sequence of formation of minerals, showing that the magnetite is mostly replaced and cut by sulfides, possibly indicating the prior precipitation of magnetite.

9.Ln 123-124: “The fluid inclusion work suggests the quartz-sulfide stage at 200-375℃ from…”The sentence is not clear.

Reply:

I am sorry that I do not know why this is not clear and I make a small modification.

The fluid inclusion work suggests the temperature of 200-375℃ at the quartz-sulfide stage [60].

10.Ln 143-144: “….digested in HNO3-HCl-HF at ~190℃….” Authors should mention the details about the grade, volume, mixing ratio etc. of the acid mixture used during sample digestion.

Reply:

Thanks.

….digested in HNO3-HCl-HF (1ml H₂[(N₃O₈)Cl] and 0.5ml HF) at ~190℃….

11.Ln 162: “….the literature values within error…..” Mention about the actual percentage of error encountered during analyses. These are very essential to assess the quality of the data presented in the manuscript.

Reply:

Thanks.

The analytical error of USGS standards is within 0.05 (2SD) and the average analysis results are consistent with the literature.

I have confidence with the quality of our data.

12.Ln 198: “…..in the order of Mag….” In this line use the same abbreviated form "Mt" for magnetite; as used in previous figures.

Reply:

I am sorry and have corrected. Thanks.

13.Ln 200: “…. δ56Fe values for mineral separates from TLS4-7 are generally lighter…..” Author should clearly explain the possible reason(s) for lighter isotopic composition in TLS4-7.

Reply:

Thanks for your suggestion.

However, we only analyzed one sample(TLS4-7) from the ore body IV and it is hard to explain the lighter isotopic composition. We hypothesize this may be related to two-stage magmatic hydrothermal event, possibly supporting the view point of ref. [53] as we discussed.

One magmatic hydrothermal event may be related to the samples from ore body III, whose ore-forming fluid has heavier Fe isotope composition. The other event is possibly concerned about the TLS4-7 and the ore-forming fluid has lighter Fe isotope composition. 

14.Ln 201: ….see above” -??? Instead of that, please refer proper table/ fig/ literature etc.

Reply:

I am sorry, it refers ref.[53] and we have decided to delete it ‘(see above)’.

Thanks.

15.Ln 217: “….is consistent with the experiment results…” What kind of experimental results ?? Mention slightly more about the types of experiment has been referred for comparison.

Reply:

Thanks for your suggestions.

The experimental Fe isotope fractionation data concerning pyrrhotite are from studies of other deposits [24,32,33] and the experimental Fe isotope fractionation factor of pyrrhotite is from ref [46].

16.Ln 220-222 “ The variation of Fe isotopic composition of pyrite is the largest, ………evolution or kinetic fractionation if any”. ---Clarify this statement with more scientific explanation.

Reply:

Thanks for your suggestions.

We have clarified accordingly as follows.

The variation of Fe isotope composition of pyrite is the largest, which may be explained using the Rayleigh fractionation (Fig. 11) or kinetic fractionation if any. Detailed discussion is given in section 5.4.

17.Ln 255-258: “The Fe isotope fractionation between chalcopyrite and pyrite is offset from equilibrium line at 350℃ and lies between the FeS-chalcopyrite equilibrium line and pyrite chalcopyrite equilibrium line at 350℃” ---This statement about Figure 8b looks fine but the related scientific explanation for this observation have to be made in detail.

Reply:

Thanks for your suggestions.

We added this explanation after the statement.

This is best understood as the FeS-pathway effect. The extent of Fe isotope exchange between pyrite and fluid controls the fluid Fe isotope composition, further affecting the chalcopyrite.

17.Ln 259-261: “In the Anqing skarn deposit, the Fe isotope fractionation ………FeS-chalcopyrite equilibrium line and pyrite-chalcopyrite equilibrium line at 350℃ may display the role of FeS-pathway effect (Figure 8b)”. But in Fig 8b there is no pyrite-chalcopyrite equilibrium line. Please check it.

Reply:

Thanks.

We have corrected this in Fig 8b.

18.Ln 263: “…..a good positive 56Fe correlation (Fig. 8c)…..” Mention about the degree of correlation in the figure 8c.

Reply:

Thanks.

We have revised accordingly in figure 8c.

19.Ln 275-279: Repetition of same text mentioned in the introduction (ln 57-62). Almost similar kind of discussion has also re-appeared later in the lines 331-340. Avoid such repetitions.

Reply:

Thanks for your suggestions.

We have revised the text as follows.

One is that the exsolved magmatic-hydrothermal fluid has a light Fe isotopic composition [40,41], in the range of δ56Fe = -0.39‰ and -0.05‰. The other [32] is that the redox state actually governs the exsolved fluid Fe isotopic composition. For example, the oxidized magmas crystallize magmatic magnetite with heavy Fe, resulting in a melt having lighter Fe and hence a lighter Fe magmatic-hydrothermal fluid. On the other hand, reduced magmas crystallize ferrous minerals (e. g. , pyroxenes and ilmenite) with light Fe, leading to a melt having heavier Fe and hence a heavier Fe magmatic-hydrothermal fluid.

Two main views about the Fe isotope fractionation during the fluid exsolution are mentioned above. One is the “light fluid” hypothesis, mainly confirmed by these studies [21-25,27,31]while a recent study [32] considers that the redox state actually governs the fluid Fe isotope composition.

In section 3 above, we discuss that the redox state of the ore fluid that governs the Fe isotopes composition of precipitated minerals [33]. In Tonglvshan deposit, precipitation of magnetite with heavy Fe will cause the fluid to become lighter, which will impart lighter Fe in subsequently precipitated chalcopyrite from this fluid. In Anqing deposit, precipitation of pyrrhotite with light Fe can cause the fluid to become heavier (Fig. 10) [27], which is expected to have great influence on the latter precipitated chalcopyrite. This is important. If we assume an initial δ56Fe value 0 ‰ for fluid, magnetite precipitation can deplete the heavier Fe, resulting in light Fe in the liquid, readily reaching a value of δ56Fe = ~ - 0.3‰ (Fig. 10). Likewise, precipitation of pyrrhotite can deplete the light Fe, resulting in heavy Fe in the liquid, readily reaching a value of δ56Fe = ~ 0.3‰ (Fig. 10).

20.Ln 284: In fig 9 caption it has mentioned that “Temperature (℃) vs. oxygen fugacity (logfO2)[27,33,59]” …..But statement doesn’t look complete. Is this representing the relation between the parameters linked to ore forming condition?? Authors should mention the details. What was the actual data source for this plot?

Reply:

Thanks for your suggestions.

Dr. Garwin first draw this diagrams during his PHD through his research about the Batu Hijau copper-gold deposit .

Dr. Garwin concluded the mineral assemblages and ore forming condition at oxidized and reduced ore deposits, respectively.

So we use this diagram to illustrate the Fe isotopes fractionation during the fluid evolution.

21.Ln 291: In figure 10a, for the Rayleigh fractionation plot, the initial isotopic ratio in fluid has assumed as 0.0‰. Authors should mention actual justification against such assumption. Why only 350C has considered as the reference temperature for those fractionation plots??

Reply:

Thanks for your questions.

The reason why we choose the 0.0 ‰ of the initial isotopic ratio in fluid in our mode is that the Fe isotope composition of the exsolved magmatic hydrothermal fluid is not clear.

There exists two point views. One study [41] suggested that the exsolved magmatic hydrothermal fluid has light Fe isotope composition estimated to be δ56Fe = -0. 39‰ to -0. 05‰, as confirmed by other studies [21-25,27,31]. However, a recent study [32] considers that the redox state actually governs the fluid Fe isotope composition: the oxidized magmas crystallize magmatic magnetite resulting in a melt with lighter Fe and hence a lighter Fe magmatic-hydrothermal fluid while reduced magmas crystallize ferrous minerals (e. g. , pyroxenes and ilmenite) leading to a melt with heavier Fe melt and hence a heavier magmatic-hydrothermal fluid.

We choose the 0.0 ‰ (“middle value”) standing for the Fe isotope values of initial fluid.

Because we referred to the hydrothermal condition from ref.[44,45] and the mineralization of sulfide stage in our study deposits is about 350℃, we choose  350℃ as the reference temperature for those fractionation plots.

22.Ln291-292: For the estimation of Rayleigh fractionation, author has used the δ 56Fefluid = δ56Fefluid (initial) 291 +103×Δ56Femineral-fluid×ln(F). Details of the data (e.g. 56Fefluid, F etc.) used in this relation should be provided separately (at least in a supplementary table with references). This will give better understanding about these plots.

Reply:

Thanks for your suggestions.

I will supplement the meaning of these variation in the text.

F is the Fe fraction in remaining fluid and the δ56Fefluid is the Fe isotope composition of the remaining fluid.

23.Ln 299-304:

“Here, magnetite from the oxidized-Tonglvshan deposit is used to calculate the equilibrium fluid Fe-isotopes composition, obtaining heaviest δ56Fe~-0.393±0.029‰ …... In the reduced-Anqing skarn deposit, we choose pyrrhotite to calculate the equilibrium fluid Fe isotopes composition and obtain lightest δ56Fe~0.045±0.057‰……within error.”

This is an important section but it needs more detailed description regarding how those optimum equilibrium isotopic ratios (i.e., -0.393±0.029 & 0.045±0.057‰) of source fluids have been estimated.

Reply:

Thanks for your suggestions.

We added the methods in the text.

δ56Fefluid = δ56Femt56FeMt-fluid,the Δ56FeMt-fluid= 0. 6‰ at 350℃

δ56Fefluid = δ56FePo56FePo-fluid,the Δ56FePo-fluid =-1. 12‰ at 350℃

24.Ln 321: At the end of section 5.3 it will be good to see some discussion on the isotopic fractionation in garnet samples. These silicate minerals have analysed for Fe-isotope ratios but in text there was no discussion on that matter.

Reply:

Thanks for your suggestions.

The reason why we do not explain the data of garnet is that the garnet in the Tonglvshan deposit has an anti-zonal structure. Previous studies have shown that the zonal structure of garnet causes iron isotope fractionation [*].

At present, our laboratory has not developed in situ analysis technology, which is the next step for our laboratory, so the garnet data will not be explained here.

25.Ln 337-340: Authors stated that the “….magnetite precipitation can deplete the heavier Fe, resulting in light Fe in the liquid, readily reaching a value of δ56Fe =~ -0.3‰. Likewise, precipitation of pyrrhotite can deplete the light Fe, resulting in heavy Fe in the liquid, readily reaching a value of δ56Fe =~0.3‰ (Fig. 10).

However in Fig 10, (which was made on model-based study); showed the Fe-isotopic variations of source fluid due to precipitation of magnetite or pyrrhotite could reach up to -2.4‰ and +2.4‰ respectively. This looks there is some mis-match between text and figure.

Reply:

Thanks for your questions.

In Fig 10a, our model demonstrates that the Fe-isotopic variations of source fluid due to precipitation of magnetite or pyrrhotite could indeed reach up to -2.4‰ and +2.4‰ respectively.

However, it is unreal because magnetite and pyrrhotite are not the dominate mineral in these two deposit and cannot precipitate ~90% Fe source.

In the text, we assume that the fluid can reach a value of δ56Fe =~ -0.3‰ and 0.3‰ because of precipitation of magnetite or pyrrhotite, respectively.

26.Ln 355: In Fig 11 the variation of isotopic ratios in pyrites and fluid have been presented. In text, authors should explain– What they meant by “Instantaneous and cumulative” pyrites? Why they considered these phases of pyrite separately for their model development or how the results of present study are linked to those phases ??

Reply:

Thanks for your suggestion.

Yes, we have explain in revision.

Instantaneous pyrite represent single pyrite formed at every instant and cumulative pyrite means bulk pyrite at every instant .

δ56Fefluid = δ56Fefluid(initial) +Δ56Femineral-fluid×ln(F)

δ56Femineral(instantaneous) = δ56Fefluid56Femineral-fluid;

F×δ56Femineral(cumulative)+ (1-F) ×δ56Fefluid= δ56Fefluid(initial) .

Because of lacking of in-situ Fe isotope analysis, We do not know for sure that the pyrite we analyzed is instantaneous pyrite or bulk pyrite. So we consder both models.

In-situ Fe isotope analysis is essential to address many detailed issues at present in our research community. We are developing the methods at present.

Reference

[*] Gerrits, A.R., Inglis, E.C., Dragovic, B. et al. Release of oxidizing fluids in subduction zones recorded by iron isotope zonation in garnet. Nat. Geosci. 12, 1029–1033 (2019). https://doi.org/10.1038/s41561-019-0471-y
